DISCOVERY REPORT

# A new human embryonic cell type associated with activity of young transposable elements allows definition of the inner cell mass

**Manvendra Singh**[1,2], **Aleksandra M. Kondrashkina**[1‡], **Thomas J. Widmann**[3‡], **Jose L. Cortes**[3‡], **Vikas Bansal**[4], **Jichang Wang**[1], **Christine Römer**[1], **Marta Garcia-Canadas**[3], **Jose L. Garcia-Perez**[3,5]*, **Laurence D. Hurst**[6‡]*, **Zsuzsanna Izsvák**[1]*

**1** Max-Delbrück-Center for Molecular Medicine in the Helmholtz Society, Berlin, Germany, **2** Max Planck Institute of Multidisciplinary Sciences, City Campus, Göttingen, Germany, **3** GENYO, Centre for Genomics and Oncological Research: Pfizer/University of Granada/Andalusian Regional Government, PTS Granada, Granada, Spain, **4** German Center for Neurodegenerative Diseases, Tübingen, Germany, **5** Institute of Genetics and Molecular Medicine (IGMM), University of Edinburgh, Crewe Road, Edinburgh, United Kingdom, **6** The Milner Centre for Evolution, Department of Life Sciences, University of Bath, Bath, United Kingdom

‡ AMK, TJW, and JLC share second authorship on this work. LDH is Lead contact author on this work.
* jlgp@genyo.es (JLGP); bssldh@bath.ac.uk (LDH); zizsvak@mdc-berlin.de (ZI)

The Editors encourage authors to publish research updates to this article type. Please follow the link in the citation below to view any related articles.

## Abstract

There remains much that we do not understand about the earliest stages of human development. On a gross level, there is evidence for apoptosis, but the nature of the affected cell types is unknown. Perhaps most importantly, the inner cell mass (ICM), from which the foetus is derived and hence of interest in reproductive health and regenerative medicine, has proven hard to define. Here, we provide a multi-method analysis of the early human embryo to resolve these issues. Single-cell analysis (on multiple independent datasets), supported by embryo visualisation, uncovers a common previously uncharacterised class of cells lacking commitment markers that segregates after embryonic gene activation (EGA) and shortly after undergo apoptosis. The discovery of this cell type allows us to clearly define their viable ontogenetic sisters, these being the cells of the ICM. While ICM is characterised by the activity of an Old non-transposing endogenous retrovirus (HERVH) that acts to suppress Young transposable elements, the new cell type, by contrast, expresses transpositionally competent Young elements and DNA-damage response genes. As the Young elements are RetroElements and the cells are excluded from the developmental process, we dub these REject cells. With these and ICM being characterised by differential mobile element activities, the human embryo may be a "selection arena" in which one group of cells selectively die, while other less damaged cells persist.

## Introduction

In broad outline, we understand early human development. From the zygote, the embryo progresses through the 2-cell stage, to 4-cell, to 8-cell (E3), to morula (E4), and thence to

**Data Availability Statement:** All data sources are noted in text or available as supplementary tables. The sources are: human pre-implantation lineages: https://www.ncbi.nlm.nih.gov/geo/query/acc.cgi?acc=GSE36552 , https://www.ebi.ac.uk/biostudies/arrayexpress/studies/E-MTAB-3929 https://ega-archive.org/studies/EGAS00001003667 Cynomolgus pre-implantation embryogenesis: https://www.ncbi.nlm.nih.gov/geo/query/acc.cgi?acc=GSE74767 Mouse blastocyst samples: https://www.ncbi.nlm.nih.gov/geo/query/acc.cgi?acc=GSE45719 https://www.ncbi.nlm.nih.gov/geo/query/acc.cgi?acc=GSE57249 ATAC-seq and RNAseq datasets from human 8-cell, bulk ICM, naïve and hESCs: https://www.ncbi.nlm.nih.gov/geo/query/acc.cgi?acc=GSE101571 ChIP-STARR-seq: https://www.ncbi.nlm.nih.gov/geo/query/acc.cgi?acc=GSE99631, https://www.ncbi.nlm.nih.gov/geo/query/acc.cgi?acc=GSE54471 https://www.ncbi.nlm.nih.gov/geo/query/acc.cgi?acc=GSE35583 All code is available from: doi.org/10.5281/zenodo.7925199.

**Funding:** Z.I. was funded by European Research Council, ERC Advanced [ERC-2011-ADG 294742]. L.D.H. is funded by European Research Council, ERC Advanced [ERC-2014-ADG 669207]. J.L.G.P´s lab is supported by CICE-FEDER-P12-CTS-2256, Plan Nacional de I+D+I 2008-2011 and 2013-2016 (FIS-FEDER-PI14/02152), PCIN-2014-115-ERA-NET NEURON II, the European Research Council (ERC-Consolidator ERC-STG-2012-309433), by an International Early Career Scientist grant from the Howard Hughes Medical Institute (IECS-55007420), by The Wellcome Trust-University of Edinburgh Institutional Strategic Support Fund (ISFF2) and by a private donation by Ms Francisca Serrano (Trading y Bolsa para Torpes, Granada, Spain). The funders had no role in study design, data collection and analysis, decision to publish, or preparation of the manuscript.

**Competing interests:** The authors have declared that no competing interests exist.

**Abbreviations:** AUC, area under curve; CCA, canonical correlation analysis; CCV, canonical correlation vectors; DEG, differentially expressed gene; DM, diffusion map; DPT, diffusion pseudotime; EGA, embryonic gene activation; EPI, epiblast; ERV, endogenous retrovirus; hESC, human embryonic stem cell; ICM, inner cell mass; IVF, in vitro fertilisation; MVG, most variable gene; NCC, not-characterised cell; NHP, nonhuman primate; PAGA, partition-based graph abstraction; PC, principal component; PCA, principal component analysis; PET, paired-end tag; PrE, primitive endoderm; PSC, pluripotent stem cell; QC, quality control; RE, retroelement; RPKM, reads

blastocyst (E5 onwards). Within the blastocyst is the inner cell mass (ICM) that gives rise to the epiblast and thence to the embryo. Single-cell transcriptomic analysis [1–4] has permitted a clearer definition of many cell types, both expected and unknown, and their ontogenetic derivatives. For example, recent analysis reveals a population of cells with trophectoderm (TE) and epiblast markers (EPI) at E6 that give rise to primitive endoderm (PrE) [3]. Nonetheless, there remains much that is not fully resolved. Perhaps most importantly, the key cell type from which the embryo is derived, the ICM, has for long while remained less well resolved in both its ontogeny and its definition ([2], see also [3,5]). Indeed, for a considerable period, ICM was thought too transitory to capture and identify [2,5]. The definition is, however, a key goal, not least because it would enable a better understanding of human pluripotency and the comparative nature of laboratory model cells, human embryonic stem cells (hESCs).

While expression of certain pluripotency genes and transcription factors all but define ICM (e.g., NANOG, GATA6), recent further characterisation additionally included apoptotic genes (e.g., in [6]), which is perhaps unexpected. More generally, programmed cell death, apoptosis, has been observed during the cleavage stage of human in vitro fertilisation (IVF) embryos (reviewed in [7]) and is common at the morula and blastocyst stages in other mammals [8]. In which cell types this occurs has yet to be discerned. Understanding the biology of such cells may allow us to understand why such apoptosis is happening.

In this context, we are especially interested in the activity of transposable elements in the early human. While about half of our genome consists of remnants of transposable elements [9], transposition of some Young retroelements (REs) (<7 MY, human–chimpanzee split), mobilised via retrotransposition, is possible [10,11]. That Young REs are transcriptionally active in early human embryos [12,13] is then intriguing. Whether this implies transpositional activity, however, is another issue. Nonetheless, to propagate to the next generation, REs must transpose either in germline or pre-germline cells. Thus, transpositional activity in the early human embryo would not be surprising. That the host evolves to suppress transposing elements [14] indicates an ongoing conflict between hosts and REs, consistent with insertion events generating both intra-clone diversity and fitness variation.

Here then, we seek to define the cell types of the early human embryo more fully, with a particular focus on the activity of transposable elements. This we do via analysis of single-cell transcriptomic data, from visualisation of early human embryos and from experiments employing hESCs. We report a population of cells (replicated in independent datasets) that does not correspond to the prior "unspecified" transitory cells [2], these having heterogeneous expression of commitment markers. This new cell type is, we find, associated with the activity of Young REs, DNA damage, and apoptosis. As these Young elements are retroelements and the cells are developmentally excluded, we dub this new type REject cells. Visualisation and single-cell analysis agree that approximately 20% of the cells of the early embryo are such cells. Having defined these, we can better characterise ICM and define marker genes. We note that Radley and colleagues [4] using a state-of-art set of algorithms, recently acknowledged that they could confirm the same marker genes as we report (presented in the early release of the present paper [15]). We additionally find that this newly defined ICM has no apoptotic features and suppresses Young transposable elements. In accord with the view that host defence systems are often controlled by co-option of other mobile elements [16], we find that an older RE (HERVH) is associated with this suppression. We discuss what the association between Young REs and REjects and Old REs and ICM might imply.

per kilobase of transcript per million reads mapped; RPPM, reads per plasmid million; SCANPY—PAGA, single cell analysis in python—partition-based graph abstraction; TE, trophectoderm; TPM, transcript per million; tSNE, t-distributed stochastic neighbour embedding.

## Results

### The developing human embryo has a population of noncommitted cells

We first sought, using single-cell transcriptome data [1,2], to classify cell types. We start by analysis of 1 dataset [2] that resolves cells prior to the epiblast stage. Using the expression of 2,000 most variable genes (MVGs) followed by unsupervised clustering, we resolve 10 clusters that we classify based on known transcriptome markers [2] (Figs 1A and S1A–S1C). Merging another dataset [1] shows that single cells cluster based on their embryonic stages and not by batch, indicating control of batch effects. In E1–E4, we straightforwardly identify clusters corresponding to oocytes, zygote, 2, 4, 8 cells stage (E3) and even the more heterogeneous morula (E4) (Fig 1A). As reported [2], human blastocyst formation initiates at E5, progresses at E6 and stabilises at E7 prior to implantation (S1A Fig). Based on their respective markers [2], we identify EPI and PrE as well as trophectoderm (polar, mural) clusters in E6 and E7 stages with resolvable markers for each [area under curve (AUC) ≥0.90] (Fig 1A). The remaining cluster from E6 and one from E6–E7 are unspecified since they express markers heterogeneously (Fig 1A), suggesting that these cells are yet to commit. One such set of cells at E6 that express TE, EPI, and PrE markers that may be the same as those recently identified as progenitors of PrE [3].

At E5, we observed 2 clusters, one expressing multiple lineage markers and one not expressing any markers of known blastocyst lineages (8-cell-morula-EPI-PrE-TE) (Figs 1A and S1B). Cells that express multiple markers are most likely transitory [2] (Figs 1A and S1B). By contrast, the cluster of cells expressing none of the known markers forms a discrete cell population. Given the absence of commitment markers, we temporarily dub them "not-characterised cells" (NCCs, Fig 1A). Partition-based graph abstraction (SCANPY-PAGA) analysis, performed on all cells of E3–E7, supports the existence of a cell cluster that is excluded from the developing blastocyst between late E4 and E5 (e.g., NCC), while transitory stages connect on the partition-based graph to the other cells of the blastocyst (S1D Fig).

To clarify whether NCCs segregate away ontogenetically from the cell lineages that have a future or are simply cells that have yet to have their future specified, we ask how NCCs developmentally relate to other cell types. We restricted our analysis to E5 cells and re-clustered them using the default features of multidimensional scRNA-seq data analysis tools, e.g., Seurat, monocle, destiny, and RNA velocity. Dimension reduction methodologies (e.g., diffusion map (DM), t-distributed stochastic neighbour embedding (tSNE) clustering, and pseudotemporal dynamics) and lineage decision trajectory (RNA velocity) at E5 project 3 distinct cell populations on developmental trajectories and indicate hierarchical branching toward either NCC or ICM (Figs 1B and 1C and S2A–S2D). With the expression of defining genes increasing gradually as cells progress towards either ICM or NCC, we can order the cells according to their "pseudotime" progression (S2C and S2D Fig). This further supports the NCC/ICM division during blastocyst formation (Figs 1B and 1C and S2A–S2F). The high diffusion distance, and the lack of directionality from RNA velocity analysis, of these 3 E5 lineages indicate that NCCs do not follow the developmental trajectory of either ICM or pre-TE after embryonic gene activation (EGA) through the progression of blastocyst formation (Figs 1B and 1C and S2A–S2D). This indicates that these are not simply cells yet to commit but are likely excluded from the lineage specification process. Indeed, in contrast to E5 transitory cells, there are no cells at E6 that resemble NCCs, consistent with their developmental exclusion (Figs 1A and S1D).

To check the replicability of our results, we repeated the analyses on an independently generated dataset from E5 [17]. Importantly, we could detect NCCs in the second E5 dataset as well, and, similar to the first dataset [2], find that they are separable from the committed cell population and marked by pro-apoptotic genes BIK and BAK expression (S3A and S3B Fig).

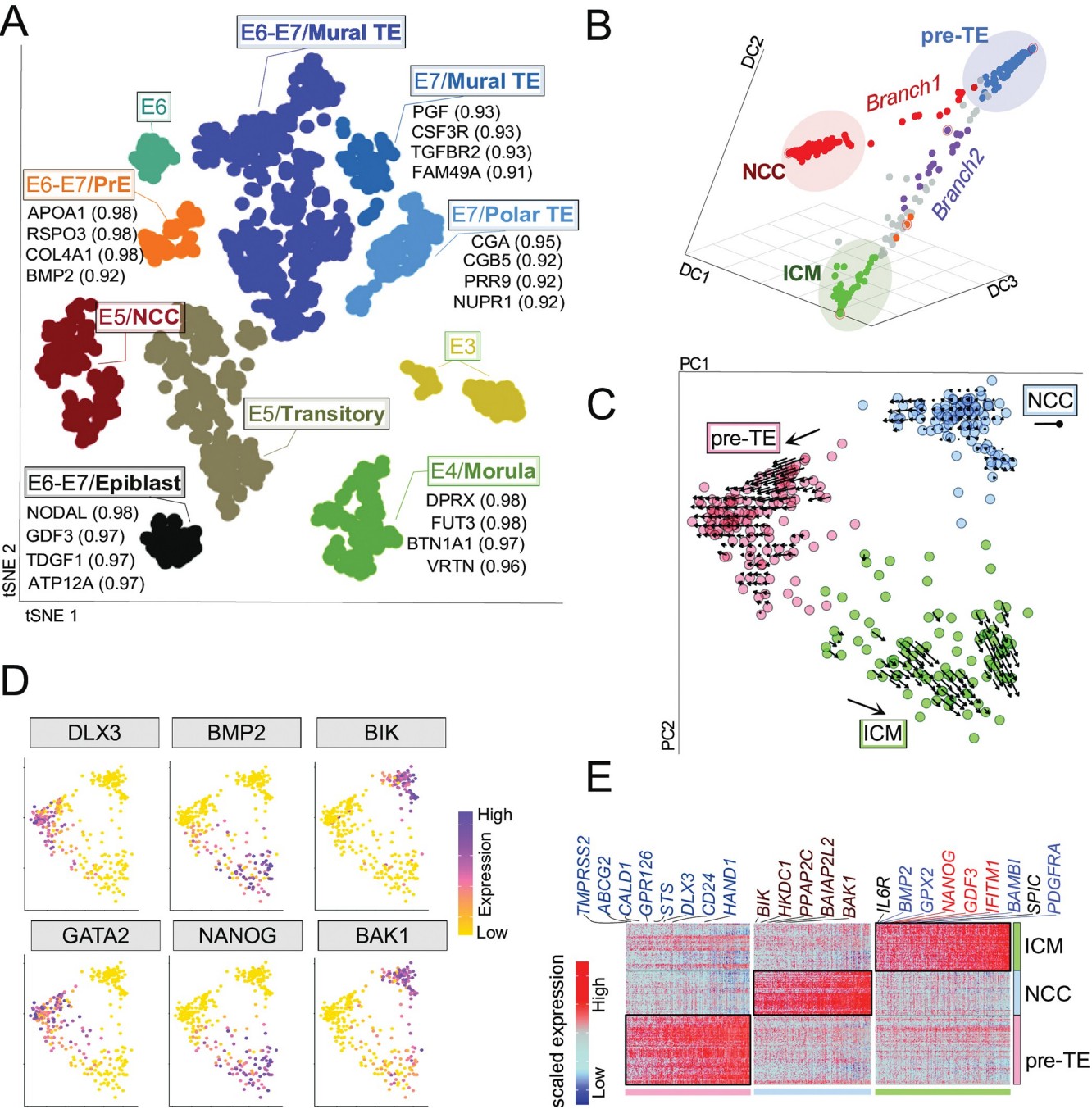

**Fig 1. High-resolution dissection of human preimplantation development identifies a NCC population.** Code to generate these figures is at doi.org/10.5281/zenodo.7925199. **(A)** Two-dimensional tSNE analysis of human single-cell preimplantation transcriptomes [1,2], using 1,651 MVGs resolves the following distinct cell populations: 8-cell at E3, morula at E4, pluripotent EPI at E6–E7, PrE at E6–E7, mural TE at E7. The E6 turquoise cluster (top left) expresses markers of TE, EPI, and PrE. At E5, cells presenting none of the known lineage markers are referred to as NCCs, whereas cells expressing multiple markers are annotated as transitory cells. For further dissection of E5, see (Fig 1B–1D). The most discriminatory genes of each cluster are listed. Numbers in brackets refer to AUC values. Colours indicate unbiased classification via graph-based clustering, where each dot represents a single cell. Note that the "E" terminology corresponds to the d.p.f. **(B)** DPT ordering of E5 cells using the top 100 MVGs (expressing (log2 TPM < 1) in 300 single cells is plotted on Diffusion Components DC1, DC2, and DC3. DPT identifies the putative branching points of NCCs (red) and ICM (purple). DM of E5 cells identifies 3 separate branches of pre-TE, ICM, and NCC. **(C)** RNA velocity projections of single E5 cells are shown on the first 2 PCs. PC biplot is showing the 3 clusters of either ICM, pre-TE, or the NCCs at E5 stage. Arrows are obtained by RNA velocity algorithms that indicate the directionality of single-cell projections. The analysis identifies NCCs as a dead-end cell population. **(D)** Feature plots based on PC plot from (C) visualising the expression of selected lineage-specific markers, e.g., NANOG, BMP2 (ICM/EPI), DLX3, GATA2 (pre-TE), BIK, BAK1 (NCC). Colour intensity gradient indicates the expression of the marker gene. Each dot represents an individual cell. Note that the NCCs are not expressing lineage markers, but are expressing BIK and BAK pre-apoptotic markers.

**(E)** Heatmap visualisation of scaled expression [log TPM] values of distinctive set of 235 genes (AUC cutoff >0.90) for each cluster shown in (C) (AUC cutoff >0.90) (for the full list of genes see S2 Table). Colour scheme is based on Z-score distribution from −2.5 (gold) to 2.5 (purple). Colour bars at the bottom highlight representative gene sets specific to the respective clusters. The ICM specific gene names in "red" or "blue" are progenitors and are also expressed at E6–E7 in EPI or PrE cells, respectively. AUC, area under curve; DM, diffusion map; d.p.f., day post fertilisation; DPT, diffusion pseudotime; EPI, epiblast; ICM, inner cell mass; MVG, most variable gene; NCC, not-characterised cell; PC, principal component; PrE, primitive endoderm; TE, trophectoderm; TPM, transcript per million; tSNE, t-distributed stochastic neighbour embedding.

While single-cell handling artefacts are more likely to result in necrosis rather than apoptosis [18], to determine whether NCCs might be artefactual, we also analysed bulk RNAseq of whole embryos (8-cell and ICM) without dissociation. Deconvolution analyses on bulk RNA-seq (GSE101571) using scRNAseq as reference detects the NCC signatures in up to 20% in the undissociated whole embryo (S3C Fig).

## NCCs overexpress DNA damage response markers and apoptotic genes

While the above suggests that NCCs have no evident developmental future, more definitive evidence for the lack of an ontogenetic future would be evidence for cell death. Compared to committed cell types, the top marker of NCCs is an apoptosis-inducing factor BIK (BCL2-Interacting Killer) (Fig 1D and 1E), accompanied by several other genes associated with programmed cell death (e.g., BAK1, BAX, and various caspases) (S3A Fig), suggesting that NCCs are likely to be eliminated from the developmental program owing to programmed cell death. For confirmatory caspase staining in human embryos see below.

The NCCs also show hallmarks of DNA damage, a likely precursor to apoptosis. As regards DNA damage, we observe, specifically in NCCs, the up-regulation of pre-apoptotic genes and multiple DNA damage response genes, including TP53I3 and TFEB [19] (S3D and S3E Fig), indicative of the activation of TP53-associated DNA damage signalling. To provide independent evidence of DNA damage, we subjected the cells to γ-H2AX/NANOG co-staining of E5 blastocyst (Figs 2A and S4A). As NANOG marks pluripotent ICM/EPI cells, we expect few cells to be stained for both NANOG and γ-H2AX. As expected, pluripotent cells stained by NANOG do not show γ-H2AX staining, while a fraction of blastocyst cells that fail to express NANOG show DNA damage (Figs 2A and S4A). These results are consistent with the possibility that cells expressing high levels of DNA damage markers are those that do not fall in the pluripotency trajectory and are prone to being excluded from the developmental process [20]. In contrast, the undamaged cells express pluripotency factors and likely to form the ICM.

## NCC identification allows an uncontaminated definition of inner cell mass

While originally thought of as too transitory to identify [2,5], the recent definition of ICM included apoptotic genes (e.g., in [6]). The above discovery of apoptotic NCCs contemporary to ICM raises the possibility that the presence of (unrecognised) NCCs may have led to an incorrect definition of ICM as having apoptotic marker genes. Indeed, the above results identify ICM as a surviving cell type of E5 embryo relatively devoid of DNA damage hallmarks.

To better define ICM, we restricted our analysis to E5, distinguishing 3 clusters on tSNE, diffusion map, and pseudotemporal trajectory (Fig 1B and 1C). Expression of DLX3 and GATA2 define [2] the first cluster as pre-TE. The second cluster, corresponding to NCC, homogeneously expresses the apoptosis inducer BAK1, the apoptosis factor BIK, but no lineage markers. The third cluster of committed cells, while expressing specific markers (e.g., IL6R, SPIC, PRR3, IFI16), co-expresses markers of both EPI (e.g., NANOG) and PrE (e.g., BMP2), and thus accords with the definition of ICM (Fig 1D and 1E). Altogether, we identified

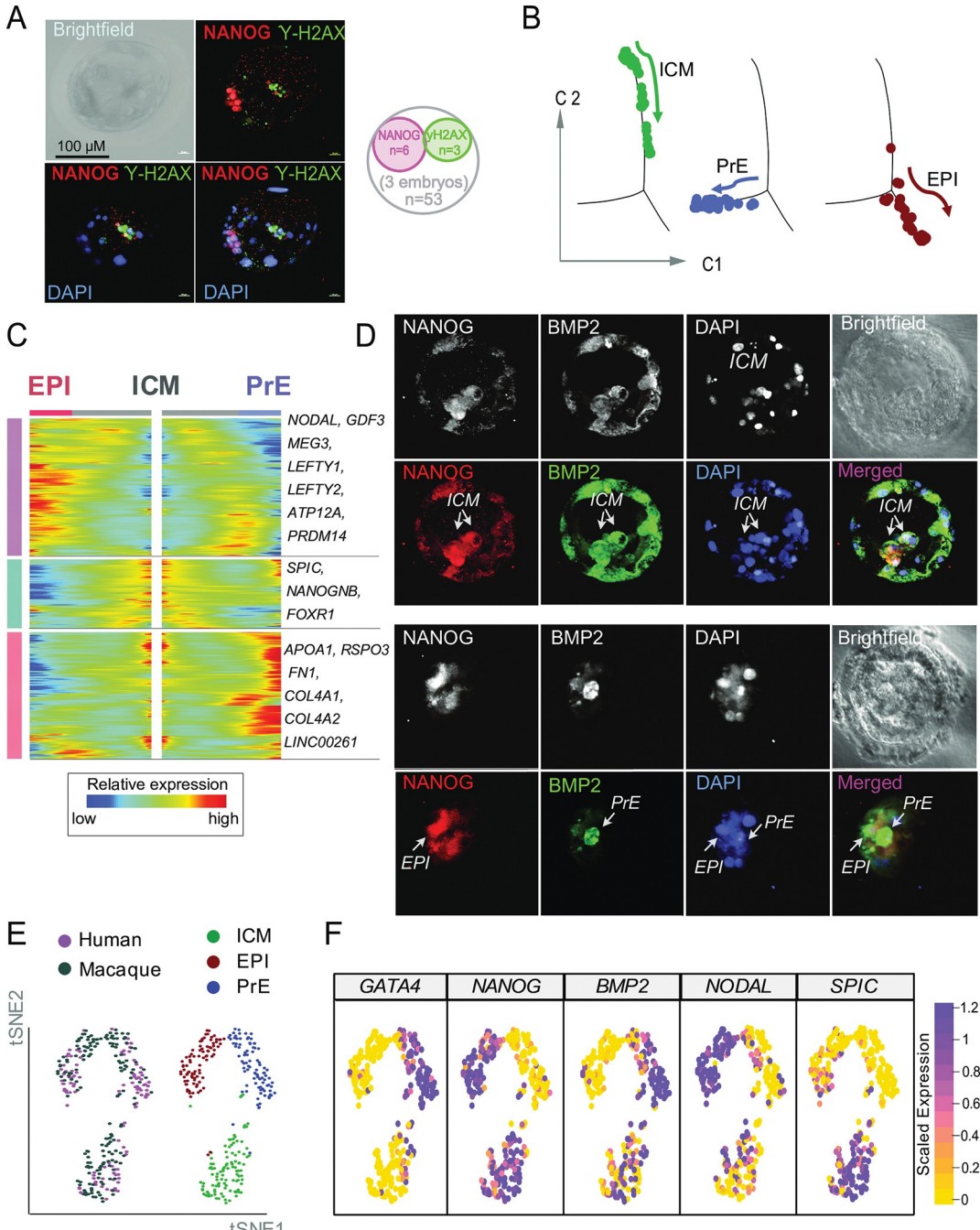

**Fig 2. Defining the human ICM.** Code to generate these figures is at doi.org/10.5281/zenodo.7925199. **(A)** Representative confocal immunofluorescent images of human E5 blastocysts co-stained against NANOG (red), γ-H2AX (green), and DAPI (blue); brightfield, black-and-white panels. Cells in the blastocyst are stained either and exclusively with NANOG, representing the compacting cells of the ICM, or with γ-H2AX, representing damaged/dying cells. Note that the γ-H2AX⁺ cells with disintegrating nucleus are distinct form the committed, pre-TE cells. A representative image is shown (for 2 more independent staining, see also S4A Fig). Magnification is 40×. Venn diagram shows the numerical analysis of the immunofluorescent co-staining performed on the 3 independent embryos (total number of cells = 53). **(B)** Monocle2 single-cell trajectory analysis and cell ordering along an artificial temporal continuum using the top 500 DEGs between ICM, EPI and PrE populations. The transcriptome from each single cell represents a pseudotime point along an artificial time vector that denotes the progression of ICM to EPI and PrE. Note that the artificial time point progression agrees with the biological time points (ICM is E5 which progresses to E6–E7 EPI and PrE). For clarity, we show this trajectory in 3 facets. **(C)** Heatmap showing the kinetics of genes changing gradually over the trajectory of ICM differentiation to EPI or PrE. Genes (row) are clustered and cells (column) are ordered according to the pseudotime progression. Genes projected in EPI are associated with

self-renewal (NODAL, GDF3, LEFTY1/Y2, CRYPTO), ICM being the progenitor lineage is marked by SPIC, NANOGNB, FOXR1, whereas, PrE projections are determined by APOA1, RSPO3, COL4A1/A2, and FN1. **(D)** Defining the human ICM as a progenitor cell population of the pluripotent EPI and the PrE. Representative confocal images (projections) of human E5early (upper 2 panels) and E5mid (lower 2 panels) blastocysts stained against NANOG (red), BMP2 (green), and DAPI, nuclear (blue) (brightfield channel (black and white)). In E5early embryos, NANOG and BMP2 co-stain cells of the ICM, whereas in E5mid embryos BMP2 and NANOG stained cells segregate, demonstrating the split of ICM into EPI(NANOG$^+$) and PrE(BMP2$^+$) cells. NCCs are expressing neither NANOG nor BMP2. Note that the nuclear aggregates of disintegrated NCCs (that would not pass the QC of scRNAseq) can be still observed at lateE5/E6 by microscopy. Magnification is 63×. (see also [29]). **(E)** Comparative single-cell transcriptomics to demonstrate ontogenetic homology to known primate *(Cynomolgus fascicularis)* [2,86] ICMs. Integrated single-cell transcriptomic data across different conditions [88] using one-to-one orthologs defined in both species. tSNE plots of ICM, EPI, and PrE cells from human (169 cells, purple) and macaque (118 cells, grey) blastocysts after the normalisation using "Seurat-alignment" (top panel). The joint clustering detects 3 distinct cross-species populations that can be identified as ICM, EPI, and PrE (bottom panel). The reclassification of the merged transcriptomes on tSNE plots reveals a similar pattern of distinct cell types in both macaque and human. Note that the macaque cells were presorted by lineage markers (e.g., ICM, EPI, PrE/hypoblast, and TE), thus no NCCs were defined. **(F)** Unsupervised identification of shared lineage markers between human and macaque. Feature plots of tSNE shown on (B), illustrate conserved gene expression in ICM, PrE, and EPI; SPIC (ICM), NANOG (ICM/EPI) and BMP2 (ICM/PrE), NODAL (EPI) and GATA4 (PrE). See S4 Table for full list of conserved markers. Note that developmental lineage modelling positions the macaque lineages at homologous ontogenetic positions in the trajectory to our defined human ICM, PrE, and EPI (see also S3 and S4 Tables). DEG, differentially expressed gene; EPI, epiblast; ICM, inner cell mass; NCC, not-characterised cell; PrE, primitive endoderm; QC, quality control; TE, trophectoderm; tSNE, t-distributed stochastic neighbour embedding.

235 marker genes (AUC >0.80) whose overexpression state are efficient molecular signatures of the individual clusters at E5 blastocyst (S2 Table).

Trajectory analysis and cell ordering along an artificial temporal continuum using the top 500 differentially expressed genes (DEGs) between ICM, EPI, and PrE populations denotes their progression (Fig 2B). The artificial time point progression agrees with the biological time points as ICM is at E5 and progresses during E6–E7 to EPI and PrE, expressing the characteristic markers (Fig 2C). To validate our pseudotime analysis of ICM diverging into EPI and PrE, we investigated the co-expression of NANOG and BMP2 in E5 and E6 embryos using immunostaining experiments (Fig 2D). Confocal microscopy of NANOG and BMP2 immunostainings at E5 showed their cellular co-expression (Figs 2D and S4A), supporting the ICM definition from our scRNA-seq analysis (Fig 2C). A few hours later, we observe that cells express either NANOG or BMP2 suggesting that EPI and PrE form as distinct lineages at around E6 stage (Fig 2D). Uncontaminated identification of the ICM is followed by lineage marker determination for the descendant lineages of the progenitor (e.g., EPI and PrE) (S4B–S4D Fig).

To further confirm our identification of ICM, we sought to clarify homology of ICM between human and primates. The same analysis also permits us to ask whether the differentiation of ICM to EPI and PrE in human is like that seen in other primates. To these ends, we performed a transcriptomic analysis of human ICM, PrE, and EPI and compared it to a primate dataset *(Cynomolgus fascicularis)* (GSE74767). Our cross-species analysis reveals the conserved nature of ICM, EPI, and PrE lineages among primates (Fig 2E) with a similar pattern of their respective marker genes' expression (Fig 2F). Overall, our confocal analyses (Figs 2D and S5A) and comparative analyses within primates (Fig 2E and 2F) support both the identification and homology of human ICM. This demonstrates that regardless of mammalian species [21], during blastocyst formation, the compacting cells near polar TE are ICM which has potential to form EPI and PEs.

## Evidence for transposition activity of Young REs and for apoptosis in the human embryo, but not in ICM

While we do not discount the possibility of multiple mechanisms of RE-driven toxicity [20,22], we hypothesised that NCCs might be affected by, or correlated with, the insertion of REs [23], as seen in oogenesis [24,25] and neurogenesis [26]. In humans, transpositionally active REs

include Long Interspersed Element class 1 (LINE-1 or L1/L1_Hs), and the non-autonomous SVA and Alu elements, mobilised by active L1 [10,11]. To examine this, we consider 2 approaches. First, we determine the expression and chromatin status of full-length "Hot" L1 elements. Second, we test for the presence of L1-encoded ORF1p protein in human embryos.

There are 89 full-length intact L1 loci in the human genome, potentially expressing both ORF1 and ORF2, required for retrotransposition [27]. To determine whether the L1 expresses both ORFs, we first calculated the coverage of RNA-seq reads over the full-length L1s in E3, E4, and E5 embryos (S5B Fig) and found the alignment of RNA-seq reads spanning their entire length, suggesting that these L1 elements express both ORFs. To determine whether the expression is driven by the L1 promoter or it is a read-through, we analysed the available ATAC-seq data from human 8-cell and ICM, as well as DNAse-seq data from morula and blastocyst [28]. We observed the significant enrichment of ATAC-seq and DNAse-seq normalised signals over the promoter sequences of full-length L1s (S5B Fig). Overall, our analysis indicates that L1s express as full-length, including both ORFs (1 and 2), and that they are likely to be driven by their canonical promoters. Six of the 89 L1 loci are activated in many cancers and are deemed as the "ultra-hot" L1 elements [10]. We identified 4 of the 6 "ultra-hot" L1 elements as a potential source of L1 activity in early human embryos (S5C Fig).

While the above analysis suggests L1 activity in morula and blastocyst, transcripts identified above may fail to provide functional translatable RNA. It is also unclear whether L1-ORFs are preferentially expressed in particular E5 blastocyst lineages. To resolve this, we performed co-immunostaining of the L1-encoded ORF1p and pluripotency factor POUF5F1/OCT4 that is expressed relatively highly in E5 blastocyst. We detect a robust expression of the ORF1p in a substantial fraction of E5 blastocyst cells (Fig 3A). As predicted, there is an inverse correlation between the expression of the ORF1p and the POUF5F1/OCT4. We find that OCT4$^+$ stained cells are not stained for L1-ORF1p. The cells that show a high intensity of POU5F1/OCT4 staining are compacted near polar TE to form probable ICM, whereas L1-ORF1p stained cells are excluded from the developing ICM (Figs 3A and S5A). In contrast to ICM, L1-ORF1p expression is readily detectable in the trophectoderm (Fig 3A and S1 Movie, see also [29]) in which the cost to the organism is lower, TE being a transient structure (see Discussion).

While large-scale transcriptional up-regulation of REs might itself trigger apoptosis [30], we asked if the DNA damage, potentially induced by L1_Hs expression (e.g., transposition of L1_Hs and SVA/Alu), might correlate with the apoptotic process in NCCs. To see if the L1-overexpressing cells are associated with apoptosis, as the single-cell transcriptomic data would suggest, we performed a co-staining with antibodies to L1-ORF1p and an early apoptotic marker cleaved Caspase 3 (cl_Caspase3) (Fig 3B). While, L1-ORF1p marks both TrEs/NCCs, and pre-apoptotic gene expression has dynamic fluctuations in the embryo, only NCCs are expected to both overexpress L1 with a fraction of them being apoptotic at any given time. Quantifying the double stained cells provides an estimate of the timing and the number of NCCs. Indeed, we observed double positive cells after morula at E5 (6/6 embryos). At E5, we observed that up to approximately 20% of cells overexpress both L1-ORFp1 and cl_Caspase3 (Fig 3B) and that apoptotic cells showed evident signs of nuclear fragmentation. The approximately 20% estimate from confocal analyses of E5 co-stained embryos accords with the approximately 20% to 30% estimate derived from transcriptome analyses.

## ICM expresses Old REs while NCC expresses Young ones

Our data indicate that ICM avoids the expression of L1_Hs elements. To see a global picture of RE expression in ICM and its the sister lineage NCC, we analysed single-cell RNAseq data [1,2] and used averaged expression (Log2 CPM+1) of each RE family across the cell

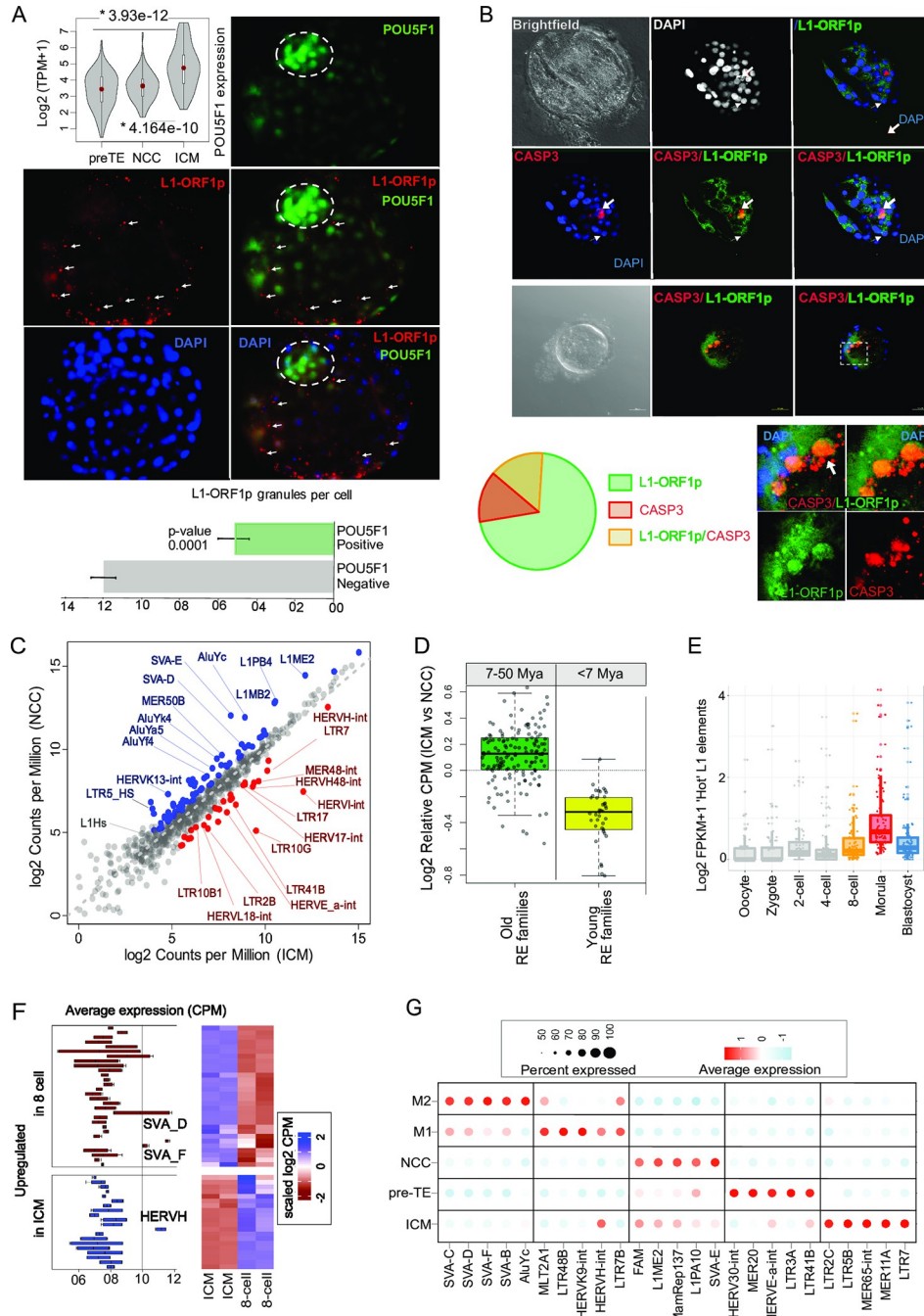

**Fig 3. LINE-1 expressing cells are excluded from the ICM.** Code to generate these figures is at doi.org/10.5281/ zenodo.7925199. (**A**) Representative confocal images show immunofluorescence staining in human early (E5) blastocysts with anti POU5F1/OCT4 (nuclear, green), L1-ORF1p (cytoplasmic granular, red), DAPI (nuclear, blue). Note: POU5F1+ cells are significantly enriched in the ICM (circled) and compacting near polar TE. A violin plot (upper left panel) visualises the density and expressional dynamics of the POU5F1 in pre-TE, NCC, and ICM at E5. Solid red dots represent the median, while quartiles are represented in the default pattern of boxplots inside the violin plots. Co-staining demonstrates the exclusive expression of POU5F1 and L1_ORF1p during the formation of blastocyst. The cells expressing higher POU5F1 compacting to form the ICM at the polar region of the blastocyst are less well stained for L1-ORF1p. L1-ORF1 stains scattered cells and pre-TE, not included in the compacted population of cells (arrows). L1 (LINE-1_Hs) belongs to a group of mutagenic, Young REs and supports transposition of both LINE-1 and the non-autonomous Alu and SVA elements. Magnification is 40×. See also S1 Movie. (Bottom panel) Numerical analysis of L1-ORF1p expression in POU5F1- vs. POU5F1+ cells in the E5 embryo. The graph shows the average number of L1-ORF1p cytoplasmic foci in POU5F1- and POU5F1+ cells, with standard deviation. Note: pre-TE

cells were not considered for this analysis. (B) Representative confocal immunofluorescent images of human E5 blastocysts co-stained against cleaved caspase 3 (cl_Caspase3) (red), L1-ORF1p (green), and DAPI (blue); (brightfield, black-and-white panels). The depicted stage III apoptotic cell overexpresses the L1-ORF1p and marked by cl_Caspase3 and has a disintegrated nucleus (Stage II—narrower arrow). Note that while the expression of pro-apoptotic markers is fluctuating in the embryo, the L1-ORF1p and cl_Caspase3 co-staining could unambiguously mark the cells that both overexpress L1-ORF1p and apoptose (specific to NCC). Two representative experiments are shown. Magnification is 63× (left embryo). The framed section is zoomed out to show the co-stained cells. The Venn diagram shows the quantification of overexpressed L1-ORF1p/cl_Caspase3 marked cells at effective E5 (data from 4 independent human embryos); 380, 296, 92, 46, total number of cells from 4 embryos, L1-ORF1p+, cl_Caspase3+, L1-ORF1p+/cl_Caspase3 +, respectively. See also [29]. Timing of the embryo is inferred from state of progression of the embryo as IVF embryos can have absolute timings different from classical. With the blastocyst still being formed, we infer this to E5 equivalent. (C) Phylogenetically young (<7 MY) and old (>7 MY) REs are antagonistically expressed in NCCs and the ICM. The scatterplot shows the comparison of normalised mean expression in CPM of various RE families between the averaged pool of ICM (x-axis) and NCC (y-axis) cells. Read counts per RE family are normalised to total mappable reads per million. Note: The top candidates are shown for both Young REs and Old REs. Young REs include LTR5_HS, AluY, SVA, L1_Hs that are human-specific and the HERVs are either specific to Hominoid or Eutherians. Uniquely mapped reads were considered as 1 alignment per read. Multimapping reads were considered as 1 alignment only if they were mapped to multiple loci, but exclusively within an RE family. Every dot corresponds to an RE family. RE families enriched in ICM (red) vs. NCC (blue). (D) Boxplot showing the distribution of averaged RE expression in ICM vs. NCCs. Note: 7 My distinguishes Old and Young REs (e.g., inserted before and after the split of human and chimp approximately 7 million years ago (Mya) [13,89]). (E) Boxplots showing the expression distribution of "Hot" L1 elements in the human embryonic development stages. Every dot represents a locus of "Hot" L1. (F) Combined boxplots and heatmaps showing distinct pattern of highly expressed transposable element families at day 3 (8-cell) and day 5 (bulk-ICM) (GSE101571) of human preimplantation embryogenesis. Note: SVA_D and HERVH-int are the most abundant REs in the transcriptome of 8-cell and bulk-ICM, respectively, and possess an opposite dynamic of expression. (G) RNA transcript intensity and density of differentially expressed RE families across the cell types of E4 and E5, following the subtraction of NCCs markers. The dot colour shows average expression and scales from blue to red, corresponding to lower and higher expression, respectively. The size of the dot is directly proportional to the percentage of cells expressing the REs in a given cell type. Note the RE expression can be considered as highly specific lineage markers. While HERVH-int is expressed both in morula (M1) and ICM, it is specifically driven by LTR7B and LTR7, respectively. CPM, counts per million; ICM, inner cell mass; NCC, not-characterised cell; RE, retroelement; TE, trophectoderm.

populations (Fig 3C). We observe a transcriptional pattern of REs that distinguishes the clusters of ICM from NCC shown on the DM (Fig 3C). We find that overexpressed REs in the NCC are relatively phylogenetically young (<7 MY, human–chimpanzee split) and include potentially mutagenic elements. In contrast, these same RE families are relatively quiet in ICM (Fig 3D).

Activated Young REs in NCCs include potentially mutagenic retrotransposons [31], such as SVA_D/E, AluY (Ya5), L1_Hs (Fig 3C). In contrast, ICM showed the overexpression of relatively older REs some of which have known regulatory activities [13,32]. The Old REs, none of which are transpositionally competent, are dominantly represented by their full-length versions: LTR2B/ERVL18, LTR41B/ERVE_a, LTR17/ERV17, LTR10/ERVI, MER48/(H) ERVH48, and LTR7/(H)ERVH in ascending order of average expression (Fig 3C). Transcriptional activation of HERVH-int in ICM coincides with the expression of the regulatory LTR7 that provides a binding platform for several pluripotency factors NANOG, POU5F1, SOX2 [33–39], suggesting a potential contribution of its regulatory activity in human pluripotency.

## RE-associated between-cell heterogeneity is identifiable at the morula stage

If NCCs are the products of between-cell heterogeneity in RE activities, we might expect to see such heterogeneity a little before NCC formation with a subclass of cells expressing Young REs. Owing to the nature of the data (pooled from different samples but ascribed the same approximate timings), we cannot say whether between-cell heterogeneities occur at the same time (contemporary) or one after the other (sequential). Given this constraint, we aim to make no strong statements about the origin of NCCs prior to E5 which would require targeted analysis.

Nonetheless, at E4 we identify 2 distinguishable cell clusters (S5D Fig). While LEUTX1 flags the 8-cell stage, the 2 clusters of human morula are marked by GATA3 and HKDC1, respectively (S5D Fig). Known blastocyst signature genes [2,40] (e.g., TFCP2L1/LBP9, ESRRB, DNMT3L) are expressed in all morula cells, but predominantly in M1 (S6A Fig). For diagnostic markers for each lineage, see (S6B Fig). Trajectory analysis including E3 to E5 allows speculation that morula-derived HKDC1 marked M2 cells may eventually accumulate in NCCs, whereas the committed M1 GATA3-positive cells are further trackable towards pre-TE (S6C Fig).

If this is an early fate decision correlated with RE expression profiles, then we expect the Young REs to be seen in the M2 population or at least within morula or earlier. Consistent with activity of Young REs somewhere in morula, we observe the up-regulation of "hot" L1_Hs elements in morula (Fig 3E). We also see the antagonistic expression of the Young ver-sus Old elements in (e.g., SVA_D/F and HERVH) between 8-cell stage and ICM (Fig 3F) indic-ative of activity of Young REs earlier rather than later in development. The M1, M2, ICM, NCC, and TE lineages are specifically marked also by the expression of distinct RE families (Fig 3G). While the most highly enriched REs are different in M2 and NCC, some Young REs are overexpressed in both M2 and NCC, such as SVA_D (Fig 3G). In both cell types, the most highly enriched REs are Young REs (both cell types have overexpressed representatives from AluY and SVA families). Old one's however also have particular expression patterns: different LTR7 variants (LTR7and LTR7B) feature in morula and ICM, respectively (Fig 3G), indicating that different subfamilies of LTR7/HERVH are expressed at different stages of early develop-ment [38]. The clusters display indistinguishable expression of G1-S-G2/M cell cycle stage markers (S6D and S6E Fig) so are not cell cycle-mediated artefacts. We conclude only that there exists a subpopulation of cells at E4 that, like NCCs, is permissive for the expression of Young REs. These observations warrant further investigations to fully decipher whether mor-ula already contains the lineage that precedes NCC.

## hESCs selected for high LTR7/HERVH expression suppress Young REs

To better understand the function, if any, of HERVH in ICM, we asked whether hESCs selected for high LTR7/HERVH activity might have an ICM-like transcriptome. If they do, then this might suggest that HERVH is pivotal to the control of the ICM transcriptome. To this end, we profiled the transcriptomes of hESCs selected for elevated LTR7/HERVH expression (HERVH-high [35]) using a GFP reporter system [35]. We calculated the expression of blastocyst lineage markers, REs, DNA damage, and apoptosis-responsive genes. As a negative comparator, we consider HERVH-depleted cells. As HERVH can regulate its transcription in trans [34,35], and can affect the expression of several HERVH loci [34,35], we use a knock down (KD) HERVH approach (S7A Fig) in pluripotent embryonic stem cells that can model the developmental stage when cells discontinue to self-renew and commit to differentiate [34,35]. We presume that dif-ference seen across the HERVH$^{high}$-HERVH$^{low}$ comparison and across the HERVH-KD versus hESC comparison tell us about the consequences of HERVH activity.

We found many genes whose expression is significantly changed in KD-HERVH/hESCs and respond reciprocally in HERVH$^{High}$/HERVH$^{Low}$ comparisons. In HERVH$^{High}$ compared to HERVH$^{Low}$, we see a down-regulation of Young REs (Fig 4A). More generally, ICM and EPI markers on the one hand and NCC, PrE-markers along with DNA damage response genes on the other, show an antagonistic pattern of expression (Figs 4B and S7B). Our top ICM markers that are reproduced in other studies [3,41], include NANOGNB, FBXL20, FOXR1, PRR13, and SPIC (Fig 4B). In our comparisons of hESCs transcriptomes, these genes are expressed (log2 TPM > 1) in HERVH$^{High}$ cells only. Similarly, the genes that mark pluripotent cells ICM/EPI (NANOG, ARGFX, etc.) and EPI-specific genes (NODAL, MEG3, etc.) are also

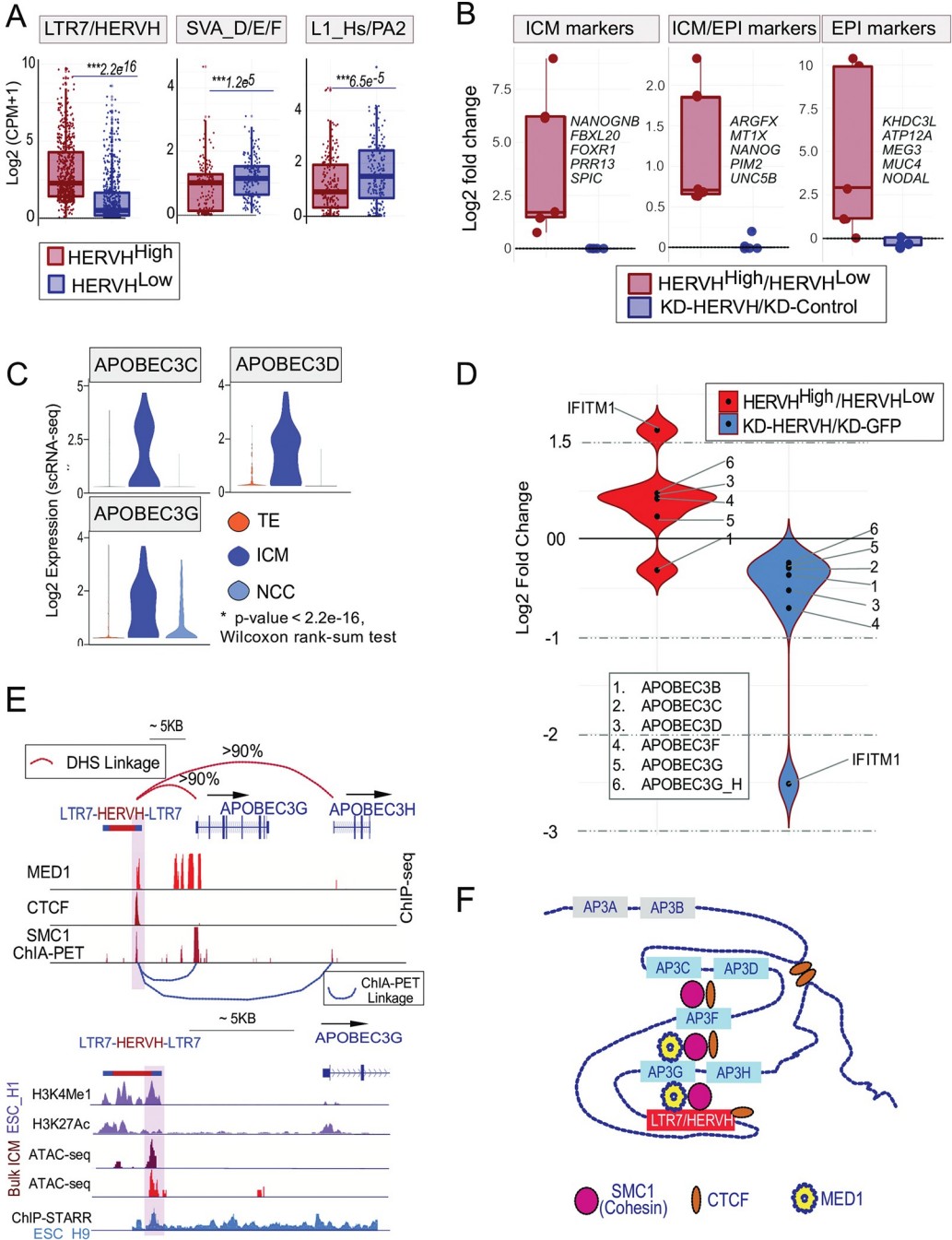

**Fig 4. HERVH regulates the expression of APOBEC3G/H in the human ICM.** Code to generate these figures is at doi. org/10.5281/zenodo.7925199. **(A)** Antagonistic expression of HERVH and Young REs in cells sorted for or against high HERVH expression (HERVH^Hig) in hESCs. Jittered boxplots show the comparison of expressed LTR7/HERVH and Young RE loci between HERVH^High (dark red) and HERVH^Low (blue) hESC_H9 cells [35] sorted for reporter (HERVH-GFP) expression. For the comparison, we considered only those loci that are full-length and expressed in either of the samples above a threshold (Log2 CPM > 1). *P*-value is calculated by Wilcoxon test. Each solid dot on the boxplot represents the expression of any individual locus of a given transposable element. Solid bold lines represent the median values, whereas the boxes are partitioned into default quartiles. **(B)** The effect of HERVH expression on lineage specification. Multiple jittered boxplots display the differential gene expression (DEG, Log2-fold change) of various blastocyst (ICM; ICM/EPI; EPI) lineage markers (red, HERVH^High vs. HERVH^Low cells; blue KD-HERVH vs. KD-GFP (control) in hESC_H1s). We show the top 5 markers. The differential expression values of the individual genes are represented by solid dots in the boxplots. **(C)** Violin plots visualise the density and expressional dynamics of APOBEC3C, 3D and 3G, implicated in host defence against REs and viruses, in NCC vs. committed cells of pre-TE and ICM. Note: The

transcription of the depicted genes mark ICM (E5, human blastocyst). **(D)** Violin plots illustrate Log2-fold changes of ICM enriched host defence genes (APOBEC3 and IFITM1) that are differentially expressed (eBayes corrected *p*-value <0.01) in the comparative transcriptome analyses; HERVH[High]/HERVH[Low] (HERVH-enriched, red) and KD-HERVH-KD/GFP-KD (HERVH-depleted, blue) in H9_ESCs [35]. Note that the HERVH affects the transcription of the APOBEC3/IFITM1 gene panel members in both conditions, but in an opposite way (*p*-value <0.00007). **(E)** HERVH as a functional enhancer of APOBEC3G. (Upper panel) IGV plot illustrating the co-occupancy of CTCF, cohesin (SMC1), and mediator 1 (MED1) signals (ChiP-seq, ChIA-PET, RNAseq) over the HERVH/APOBEC3G locus. Significant SMC1-ChIA-PET linkages are shown as blue chain lines. Red chain lines are showing high-confidence correlations of DNase Hypersensitive Sequences (DHS linkage) between the HERVH enhancer and the APOBEC3G/H genes from 79 cell lines. Genome browser view showing H3K4Me1, H3K27Ac, ChIP-STARR-seq [45] (hESC_H9), ATAC-seq (2 replicates of freshly isolated human bulk-ICM) [44] signals at the APOBEC3G locus, including the upstream full-length LTR7-HERVH-LTR7 in human PSCs. Shadowed region highlights the overlapping peaks at the HERVH. **(F)** Schematic representation of the interaction domain at the HERVH/APOBEC3(AP3) locus based on merged analyses of Hi-C, ChIA-PET, and ChIP-seq datasets. Note that while APOBEC3C/D/F/G/H are in a same domain with HERVH, APOBEC3A and B are located in a separate domain. The domains borders are marked by CTCF binding motifs. The MED1 signal marks potential super-enhancers. DEG, differentially expressed gene; EPI, epiblast; hESC, human embryonic stem cell; ICM, inner cell mass; NCC, not-characterised cell; PSC, pluripotent stem cell; RE, retroelement; TE, trophectoderm.

up-regulated in HERVH[High] cells (Fig 4B). Collectively, our data suggest that hESCs selected with high HERVH activity are enriched with ICM and EPI markers and hence accord with in vivo human pluripotency, i.e., HERVH[High] are ICM-like cells.

Upon HERVH depletion, analysis of the differential transcriptomes along with the up-regulation of Young REs reveals the activation of DNA damage sensors (e.g., TP53I3, TP53I11/13) and apoptotic factors (e.g., CASP3/7, BOK) (S7C Fig). In contrast, the expression of telomere maintaining genes (e.g., TERT) is down-regulated, and the telomeres are shortened (S7D and S7E), consistent with a role of LTR7/HERVH in the regulation of genome stability. The activation of Young REs on HERVH knockdown and their suppression in HERVH[High] cells indicate that copies of LTR7/HERVH, in addition to their regulatory role in pluripotency [42], might have been recruited as part of a repression mechanism targeting Young REs. Note that the KD cells are not the same as HERVH[Low] cells, the latter being primed hESCs, and there are no mimics of the NCC cells in ESC populations. Nonetheless, cells can be generated that capture some features of NCCs (RE up-regulation) by knocking down HERVH in ESCs.

## LTR7/HERVH induces APOBEC3 expression in ICM, suppressing Young REs

Old (e.g., HERVH [33,34,36]) and Young REs (e.g., SVA_D/E) show antagonistic expression patterns, the latter being down-regulated in ICM (Fig 3C and 3D). In fact, we do not detect L1_Hs expression in ICM (Fig 3A and 3C). Might then ICM be expressing a group of host factors that restricts retrotransposition? Consistent with this, upon re-surveying the E5 transcriptomes, we find that of the all the earliest embryonic cell types expression of RE-restricting factors (APOBEC3C, D, G) [34,35] is detectable in ICM exclusively (Fig 4C).

As hESCs selected for high LTR7/HERVH express ICM markers that are down-regulated upon HERVH depletion, we ask whether the expression pattern of APOBECs and Young REs follows these changes reciprocally. Indeed, we find that the expression of APOBEC3C/D/G and IFITM1 (an endogenous retrovirus suppressor in human ESCs [43]) changes along with LTR7/HERVH, and both have an expression profile that negatively correlates with the transcription of Young REs (Fig 4D). These results are consistent with the hypothesis that the expression of HERVH in ICM is associated with the up-regulation of Young RE suppressors.

While we do not wish to fully decipher the mechanism of how LTR7/HERVH suppresses Young RE activity, we note the relative up-regulation of APOBEC3G located downstream (10 kb window) of a full-length LTR7/HERVH in ICM cells (S8A and S8B Fig). To validate a

potential *cis*-regulatory effect of this particular LTR7/HERVH copy, we analyse ATAC-seq [44,45] and STARR-seq [45] datasets in the corresponding genomic region. We find an accessible chromatin region (ATAC-seq) overlapping with a functional enhancer signal (STARR-seq) [45] (Fig 4E) over this full-length LTR7/HERVH element located upstream of APO-BEC3G. As transcriptionally active HERVH is implicated in defining 3D structure of the chromatin in human pluripotency [32], we also analysed Hi-C [32] and ChIA-PET of cohesin coupled with PolII, CTCF, and MED1 [46] data. This approach reveals physical looping of this LTR7/HERVH with the promoters of APOBEC3G and APOBEC3H, mediated by cohesin (Figs 4F and S8C). Collectively, these data together with a high DHS co-segregation value (rho > 0.90), determined from 79 pluripotent stem cell (PSC) lines, provides evidence that LTR7/HERVH element is a co-opted enhancer of APOBEC3G/H and is fixed in humans (S7F Fig).

It is important to note that pre-TE has a lower level of expression of HERVH and thus APOBEC3s (S8D Fig), suggesting that the ability of efficiently suppressing Young REs might be specific to ICM. In contrast to ICM, pre-TE cells appear to tolerate some Young RE activity. This accords with the observed expression of L1_Hs in morula and our finding of the L1_Hs-ORF1p[+] cells are outside of ICM but in the pre-TE lineage (Figs 3C and S6B and S1 Movie) (see also [29]).

## Discussion

Here, we have provided a high-resolution view of the human embryo that has clarified numerous issues. First, we have shown, by multiple approaches, evidence for a common cell-type in the human early embryo that segregates shortly after EGA and that expresses Young REs, DNA-damage response genes, apoptotic factors, and no lineage-specific markers (see S9A Fig). The apoptotic status, expression of Young REs and DNA damage response are seen both in scRNA data and through staining of human embryos. As the Young elements are REs, we dub this new cell type without a developmental future REject cells. Second, resolution of these has permitted a clearer characterisation of the human ICM, highlighting the role of HERVH in suppressing Young transpositionally active REs. Presence of apoptotic markers in prior analysis of ICM (e.g., in [6]) most likely reflects REject contamination. Third, previously unspecified cells appear to be transitional cells.

Further, while not a focus of our analysis, our data clarifies that human development from ICM to EPI/PrE is sequential rather than simultaneous. Consistent with observation in mice [21], we find that within ICM of human and nonhuman primates (NHPs), both EPI and PrE defining genes are co-expressed. Our computational modelling of subsequent ICM divergence into EPI and PrE reports a similar pseudotemporal trajectory to that seen in *Cynomolgus* suggesting both the conservation of EPI/PrE specification and that human ICM is a distinct lineage that specifies EPI and PrE as sequential lineage bifurcations. This rejects the hypothesis of simultaneous blastocyst lineage specification in human early embryos [2]. Since we first reported this [15], the finding has been replicated in recent studies, via diverse methods [3,41].

### HERVH as a "gun for hire"

Our data also suggests a role for an Old RE, HERVH, in ICM suppressing Young REs. There is no reason to suppose that HERVH is the unique source of Young RE suppression, nor that HERVH controls Young REs only via APOBEC3s. Indeed, how APOBEC3, a potent inhibitor of retroviruses [47], might modulate Young RE levels observed within the transcriptome is not immediately apparent. Originally considered to have these effects via DNA/RNA editing [48], its main antiviral activity is now thought to be owing to the blocking of reverse transcription,

independent of any editing effect [48]. We suggest that such a process acting on Young REs such as L1 with reverse transcription abilities is worthy of scrutiny.

The observation that HERVH appears to be involved in the suppression of Young REs fits well into the recently observed trend that anti-mobile element and antiviral defence systems often employ resources, such as site specific nucleases, derived from other mobile elements, the so-called "guns for hire" hypothesis [16], CRISPR-Cas9 being a case in point. One possible reason for the employment of recently acquired mobile elements is that, as host defence is intrinsically costly, selection will favour loss of a defence system once the parasite has been suppressed and eliminated resulting in a cycle of loss then need [16]. In such a context, employing genes that can function for host defence is a more likely means to assemble a novel defence system [16]. With mobile elements also having "tools" that are fit for purpose (the guns for hire), co-opting mobile elements to the defence against other mobile elements may then be a regular means to evolve new costly host defence systems against novel parasites and mobile elements. Domestication of RAG transposase for V(D)J recombination is an exemplar. Our data add a further exemplar and suggest that co-option of endogenous retroviruses (ERVs) in mammals is a viable means to recruit novel defence mechanisms. In our case, HERVH enhances the expression of APOBEC3. The characteristic of this old RE that makes it fit for purpose is its ability to act at a chromatin level, providing a platform for binding of transcription factors that are operational in germline and pre-germline.

## Are REject cells primate specific?

REject cells are unlikely to be unique to humans. Indeed, blastomeres exhibiting extensive DNA damage (marked by γ-H2AX+) that are prevented from incorporation into blastocysts were recently reported in an NHP [49]. By contrast, in repeating the above single-cell analysis on data from mice, we find no evidence for REjects in the same time frame as seen in human data (S10A–S10D Fig). While prima facie this does not support the presence of REjects in mice this comes with a major caveat, this being that at the important blastocyst stage there are relatively few cells in the murine sample [21]. It would require deeper analysis to robustly confirm or reject their existence. This being said, while in human morula, we identify 2 cell types (S5C Fig), one associated with Young RE expression, we find very little overlap of gene expression (approximately 10%) between mouse and human morula (where REjects might be routed back) and no evidence for heterogeneity in mouse morula (S10C Fig).

An alternative reason for any possible mouse–human difference is that REject identification in primates is an artefact. The human embryonic evidence above is all derived using in vitro cultured embryos. As such, it is not possible to exclude the possibility that the observed apoptosis-mediated elimination of REjects from the human ICM is an in vitro culturing artefact. However, there are several arguments against their being artefacts of either culturing or scRNA analysis: culturing damage might predict necrosis not apoptosis; if an artefact, it is a replicable one, REjects being detected in independently generated datasets at the same time point; as above, what may well be REjects are also observable in primates [49]; deconvolutions and immunostaining experiments argue against their being an artefact of single cell handling, and, finally, their existence accords with prior data (reviewed in [7]). We encourage further in-depth single-cell analyses to both confirm existence in humans and to ascertain presence/absence in other taxa.

## Is the human embryo a clonal selection arena?

While we have identified a set of cells that are dying and that are associated with the activity of Young REs, we have not addressed the question of why this association is seen. There are

several possibilities that can be broadly classified as (i) REs being causative of the trajectory to cell death; or (ii) RE expression happening because cells are being directed towards apoptosis.

One possibly is that RE activity is causative of REject status because they induce damage. In this model, REjects are part of a quality control (QC) mechanism. This would be consistent with the observation that RE insertion can trigger DNA damage that in turn activates apoptosis [50–52]. Such QC could be mediated either via RE insertion being directly harmful (e.g., insertion into an essential gene) or through the cell sensing insertion (e.g., DNA damage response). In the latter case, cells are preemptively killed for which parallels from virus biology are well described [53]. As with retroviral insertion [54], RE activity might trigger DNA damage that activates apoptosis [50,51]. We have some evidence for this, at least as a plausible explanation. To determine whether L1 activity could activate apoptosis in cells with reduced HERVH expression, we inhibit L1 retrotransposition with capsaicin [55] in KD-HERVH cells and monitor DNA damage (S9B Fig) and early-stage apoptotic signals. We observe a significantly reduced level of both damage (γ-H2AX) and apoptosis (cl_Caspase3) signals in KD-HERVH cells treated with the inhibitor (S10C–S10D Fig), supporting the hypothesis that L1 retrotransposition could contribute to DNA damage and apoptosis in HERVH-depleted cells. Whether this reflects what is happening in early embryos is, however, unknown.

This QC model, suggests that the human embryo is a so-called clonal "selection arena" [56] in which, owing to heterogeneities that developed within the clone [56], some cells are selectively removed. Were transposable elements at least partially causative, then the human embryo would resemble oogenesis [24,25] and neurogenesis [26], in which variation owing to the jumping of REs creates the context for selection within a clonal population of cells.

The idea that the human embryo might be a selection arena may not be as outlandish as it might at first appear. Intriguingly, induced expression variation of a single gene between genetically identical cells can result in between-cell competition, leading to apoptotic cell death of some cells [57]. If it is a selection arena, there is, however, no reason to suppose that RE activity is the sole cause of between-cell variation. Human preimplantation embryos can display chromosome mosaicism, represented by euploid and aneuploid cells [58], but aneuploid cells are rarely observed in the human ICM, suggesting that they are selected out [59,60]. In line with this, analysis of primate preimplantation embryos revealed the elimination of chromosomally abnormal (aneuploid mosaic) blastomeres [49]. Segregation of heteroplasmic cytoplasmic variants may have a role in mammalian oocyte atresia [61] and possibly also in early development [62]. Point mutation, however, is unlikely to generate much variation as the rate in human germline is only approximately 0.06 per genome per cell division [63], of which only approximately 10% are under selective constraint [64]. By contrast, RE insertions or activity anywhere in the genome could potentially trigger a damage response.

A further possibility is that REs are causative of cell type specification but not because they are harmful. While the genomic loci of Young REs are typically inactivated during the morula-ICM transition, copies of certain Young REs (e.g., SVA, LTR5_Hs) might have been co-opted to have a regulatory function following EGA. Indeed, multiple loci of SVA and LTR5_Hs have open chromatin status in morula [65,66]. In mouse, L1 expression is important for proper preimplantation embryogenesis [67,68], arguing against a necessarily mutagenic activity of REs in early development. Indeed, L1 RNA recruits repressors to exit a transcriptional program specific to the 2-cell (2C) embryo and promotes gene expression associated with the subsequent developmental program [68]. This function of L1 RNA does not require the RNA to be translated. While it is possible that, as in mice, L1 (or perhaps other REs), regulates transition between developmental stages in human preimplantation development, this has yet to be shown.

Alternatively, REjects might overexpress mutagenic REs because they have no developmental future (rather than having no developmental future because they express damaging Young REs). If a cell type is defined by up-regulation of apoptosis genes to have no developmental future, possibly as they are surplus to requirement or in the wrong place, then we expect relaxed controls on gene expression, unregulated expression of Young REs being one such manifestation. Alternatively, cells may be committed to die and Young RE up-regulation may be a means to achieve this. In this model, they are both consequence (of being in an unwanted cell population) and cause (of cell death).

Understanding why REjects have Young RE expression while ICM express Old REs is then a complex problem of cause and effect. We have no unambiguous evidence here. One might note that the timing of events argues for a RE causative model: As the Young REs are the first to express after EGA, they are unlikely to be a consequence of REject formation, this possibly happening at the earliest in morula. This suggests that Young RE expression does not reflect relaxed gene expression control in cells already destined to die. The very early expression of the Young REs in turn also suggests that their expression may be contingent on a lack of effective host control/suppression soon after EGA. Similarly, apoptosis post-dates Young RE expression, implying that Young RE expression in blastocyst is not a response to apoptosis but maybe the lack of efficient RE suppression (e.g., APOBEC3). While the temporal data is suggestive, direct demonstration of RE insertion in the embryo, damage and subsequent apoptosis would be of value. Likewise, removal of the system of RE suppression should result in more REjects—and possibly embryonic lethality—if RE insertion is causative.

In addition to these temporal concerns, from an "opportunity" point of view deleterious activity of Young REs would be expected shortly after EGA. To control Young REs, we have evolved a detection and suppression system. However, while H2A.X both regulates gene activation and provides some level of Young RE regulation [69] at EGA, Young REs, could take the window of opportunity immediately after EGA to express and transpose before silencing mechanisms come fully into operation. These counter-acting processes could then quite probably generate within-clone, between-cell heterogeneity such that some cells are more heavily affected by RE damage than others. If this is right, those cells that avoided and/or managed to control Young RE expression/activity (e.g., expressing APOBEC3) are permitted into ICM formation. By contrast, the excessively damaged cells are routed into a developmental dead-end (REjects) or permitted to form the ephemeral trophoblast. Part of this contest to suppress Young REs involves an Old RE (HERVH) which, as previously established [34,35], is also needed for pluripotency specification.

One problem with this QC model is that TE also has L1 expression but without apoptosis. We do not, however, necessarily expect perfect concordance between Young RE activity and apoptosis if only because the level of QC stringency is expected to differ between cells. For example, cells of the extra-embryonic tissues have a limited future existence and so may tolerate transpositional events more than cells of the embryo. The same logic explains differential mutation rates between plant tissues predicted by the longevity of the tissues [70] and between germline and soma in mammals [63,71–73]. Indeed, recently, human placenta has been shown to have a high mutation rate and appears to be a dumping ground for aneuploid cells, consistent with a similar selection arena/QC model for early embryogenesis [74]. In a companion paper [29], we observe L1 expression in trophectoderm but, in contrast to REjects, these cells do not express pre-apoptotic factors, suggesting a certain tolerance toward L1 activity. The companion paper [29] provides evidence both for transposition in vivo and for a lower rate of recovery of new RE insertions in viable human progeny than observed in early embryogenesis, consistent with a selection arena.

## Implications

Beyond the fact that that resolution of REjects allows uncontaminated definition of human ICM, what are the implications of our results? We highlight 2 areas: fertility and ethics. As a surge of RE activity might determine the fate of the embryo, not just a limited set of cells, our results have potential implications for possible causes of infertility. While some RE activity is tolerated, resulting in RE integration in the trophoblast (see also [29]) and even in heritable RE insertions accumulated through transposition at this stage of human development [75], if the selection arena model is correct, then activity of mutagenic REs could result in the embryo (and not just REjects) being selected out. This may be owing either to REs being particularly active or through oversensitivity to their activity. In the latter regard, perhaps significantly, a p53 non-synonymous polymorphism is associated both with apoptosis potential and with recurrent pregnancy loss [76,77]. The involvement of p53 is also notable as it has an intimate relationship with Young L1s, and can drive a positive feedback loop to amplify L1 retrotransposition, resulting in apoptosis [78]. As it can indeed induce differentiation of pluripotent stem cells or apoptosis [79], p53 is a likely candidate to play a gatekeeper function to ensure high fidelity development.

The discovery of a new cell type defined by (we presume) chance expression of Young REs also questions assumptions made by both embryologists and ethicists. For example, Austriaco [80] assumes that "an organism. . . is a deterministic system that follows a particular developmental trajectory." He argues that this deterministic perspective "can go far in clarifying many of the controversies in contemporary biomedical ethics." From this perspective, the born embryo is the determined product of the fertilised egg undergoing embryogenesis and so the 2 should have equivalent moral status. The idea that the early embryogenesis is not strictly deterministic with cells differentially affected by potentially stochastic RE activity, may then be oblige reevaluation of such arguments.

## Methods

### Data selection

We (re)analysed datasets from 11 different studies. In order to dissect the human preimplantation lineages, we re-analysed single-cell RNAseq datasets from *Homo sapiens* (GSE36552, E-MTAB-3929 and EGAS00001003667) and *Cynomolgus* preimplantation embryogenesis (GSE74767). We used predefined ICM, EPI, PrE, and TE samples from *Cynomolgus*, in order to perform comparative analysis with their human counterparts. We performed comparative studies using mouse blastocyst single-cell transcriptome samples (GSE45719 and GSE57249) to identify NCCs and the self-renewal network in EPI. We used ATAC-seq and RNAseq datasets from human 8-cell, bulk ICM (ICM/NCC mix), naïve and hESCs (GSE101571) and ChIP-STARR-seq, (GSE99631, GSE54471, and GSE35583) to decipher the HERVH-target genes in human pluripotency. ChIA-PET of cohesin (SMC1), ChIP-seq of CTCF, MED1 and K27Ac in human pluripotent cells were reanalysed from GSE69647. The Hi-C data was from GSE116862.

### Clustering strategy

We employ a strategy of clustering MVGs from whole cell transcriptomes, using a combination of clustering K-means and principal components (PCs) [81]. We identified 1,597 genes exhibiting high variability potentially useful for defining cell types (Fig 1A). To reduce data dimensionality, we performed principal component analysis (PCA) and enlisted the significant

PCs using a "jackstraw" method [81]. This identified 9 significant PCs ($p$-value $<10^{10}$), these being employed as inputs to tSNE for visualisation.

### Single-cell RNAseq data processing

Activity of genes in every sample was calculated at TPM expression levels. We considered samples expressing more than 5,000 genes with expression level exceeding the defined threshold (Log2 TPM >1). We considered genes expressing in at least 1% of total samples for the analysis. This resulted in 1,285 single cells of human E3–E7 samples, with 15,501 expressed genes. We used Seurat 1.2.1 for clustering E3–E7 and E5 cells (this version was the latest during the preparation of the manuscript), whereas the rest of analysis was carried out using Seurat_2.2.1 (http://satijalab.org/seurat/) packages from R to robustly normalise the datasets at logarithmic scale using "*scale.factor = 10000*". After normalisation, we calculated scaled expression (z-scores for each gene) for downstream dimension reduction. The cells were separated by subjecting the MVGs ([log(Variance) and log(Average Expression)] > 2) to the dimension reduction methods of PCA.

### Principal component analysis (**PCA**)

As previously described [82], we ran PCA using the "*prcomp*" function in R, then utilised a modified randomization approach ("*jack straw*"), a built-in function in the "Seurat" package, to identify "statistically significant" PCs in the dataset. We used the genes contributing to the top 9 significant PCs for E3–E7 stages, 5 significant PCs for E3–E5 stages, 3 significant PCs for E3–E4, and the first 2 significant PCs for E5 stage as input to visualise in 2D with tSNE.

### t-Stochastic neighbour embedding (tSNE)

Using the above significant PCs as an input, we applied *tSNE*, a machine learning algorithm to cluster the cells in 2 dimensions. To define cell population clusters, we employed the *FindClusters* function of "Seurat" using "PCA" as a reduction method. To resolve the clusters on tSNE, the density parameter {G.use} was set between 6 to 10, and the parameter providing the fewest clusters was chosen for visualisation. This approach identified 10 clusters from E3–E7, 5 clusters from E3–E5, 3 clusters from E3–E4 and E5 stages. The specific markers for each cluster identified by "Seurat" were determined by the "FindAllMarkers" function, using "roc" as a test for significance. A gene matching the following criteria was considered as a marker for a given cluster: (i) the gene is overexpressed in that particular cluster (average fold difference >2 compared to the rest of the clusters); (ii) the gene is also expressed (Log2TPM >2) in at least 70% of the cells in that particular cluster; and (iii) the AUC value is greater than 80%.

Feature plots, violin plots, and heatmaps were constructed using default functions, excepting the colour scale that was set manually. The annotated ICM-EPI-PrE cells were re-clustered using the methodologies described above and visualised on the tSNE coordinates.

### Trajectory and diffusion analysis

Trajectory analysis of the differentiation process fromprogenitors to committed cells was performed using the Monocle2package [83], which generates a pseudotime plot that graphically illustrates the branched and linear differentiation processes. For pseudotemporal analysis of human and macaque data, we first imported the processed Seurat object into the Monocle2 workspace using "importCDS" function. The datasets were processed further using the series of default functions with negative binomial expression family parameters. The final dimensionality was reduced to 2 components. The dimensionality of the data was reduced by

constructing a parsimonious tree using "DDRTree." We employed the "differentialGeneTEst" function to find the top DEGs (q-value $< 1 \times 10^{-8}$), these being fed as an input for unsupervised ordering of the combined set of cells. Note: The q-value threshold was set manually ($10^{-8}$–$10^{-15}$) to find the root, branching point and leaves on the trajectory graphs. This approach identified the genes (100 to 1,000) that were significantly differentially expressed. We then used the expression data of these genes to construct a DM for the respective cell populations (DiffusionMap function in the Destiny package [84]). We calculated the diffusion pseudotime (DPT function in the Destiny package). Finally, the cells on the DMs and trajectories were annotated on the basis of their identity, previously determined from the Seurat analysis. A similar strategy was also used on data from E5, E3–E5, and ICM-EPI-PrE cells. Genes were plotted on a branched heatmap obtained by setting a threshold of differential expression to qval $< 1 \times 10^{-8}$.

## SCANPY-PAGA analysis

In order to confirm the existence of NCCs in the early human embryo, we performed single cell analysis in python—partition-based graph abstraction (SCANPY—PAGA) analysis from single cell (sc)RNAseq data. We used raw counts (E-MTAB-3929) for the partition-based graph abstraction (PAGA) analysis. First, we annotated the scRNAseq data with the given features (Embryonic days, Embryonic stages, Embryo ID, and genes) using R packages "SingleCellExperiment" and "scRNAseq." The obtained R object was converted into "loom" format, which was imported into SCANPY. We used default parameters of SCANPY for preprocessing and visualisation. Clustering (PAGA) was performed using "Louvain" method, where the threshold for resolution was kept at 0.1. The plotted clusters agreed with the results we obtained from Seurat.

## Deconvolution analysis

We used Bseq-Sc [85] for the deconvolution analysis. Data of bulk RNAseq of 8-cell and ICM (GSE101571) were subjected to the analysis, using the obtained clusters from single cell (sc) RNAseq as a reference. We first constructed the "Expressionset" for both datasets and then performed the reference-based deconvolutions using default parameters.

## Analysis of repetitive elements

To estimate expression levels for repetitive elements, we used 2 strategies. The longer reads in Yan and colleagues' data [1] allowed us to calculate CPM or RPKM for individual RE loci. In contrast, data from [2] was suitable only to detect the average expression of any given RE family as it was unable to unambiguously map exclusively to any given locus. In this instance, we considered multimapping reads only if they were mapping exclusively within an RE family. One alignment per read was employed to calculate counts per million (counts normalised per million of total reads mappable on the human genome). The expression level of repeat families was calculated as Log2 (CPM+1) prior to comparison. SVA_D elements from NCC and HERVH-int elements from EPI were the most abundant in the respective cell types. Given this, we removed the very few single cells that showed extremely low or non-detectible expression of these in the relevant cell type (Log (CPM+1) < 0.1). Note that datasets from different layouts (single versus bulk RNAseq) were never merged into 1 data frame to perform REs comparative analysis, not least because we were unable to normalise these datasets.

## ATAC-seq data analysis

ATAC-seq raw datasets in sra format were downloaded and converted to fastq format using *sra-tools* function *fastq-dump–split-3*. Fastq reads were mapped against the hg19 reference genome with the bowtie2 parameters:—very-sensitive-local. All unmapped reads, reads with MAPQ < 10 and PCR duplicates were removed using Picard and *samtools*. All the ATAC-seq peaks were called by MACS2 with the parameters—nomodel -q 0.01 -B. Blacklisted regions were excluded from called peaks (https://www.encodeproject.org/annotations/ENCSR636HFF/). To generate a set of unique peaks, we merged ATAC-seq peaks using the *mergeBed* function from bedtools, where the distance between peaks was less than 50 base pairs. We then intersected these peak sets with repeat elements from hg19 repeat-masked coordinates using bedtools *intersectBed* with 50% overlap.

## Identifying HERVH target genes

We determined HERVH-derived functional enhancers in human pluripotent cultured stem cells (hPSCs) by data mining the merged analysis of ChIP-seq, plasmid DNA-seq, and ChIP-STARR RNA-seq [45] (GSE99631). Using these genome-wide analyses, we first collected HERVH loci with RPPM >144 reads per plasmid million (RPPM). This strategy identified a list of 543 distinct HERVH loci as functional enhancers in hPSCs (STARR-seq was performed on primed cell types [45]). In addition, we aimed at determining open chromatin regions around HERVH loci from bulk ICM cells (note: this was a mixture of ICM and NCCs) [44]. First, the signal file (wig format) of ATAC-seq in ICM and hESCs was downloaded and RPKM values were calculated per 100 bp-window. As previously [44], windows carrying RPKM >2 were considered as open chromatin regions. Next, we intersected these regions with the 100 bp bins of distinct full-length HERVH loci, resulting in a list of HERVH loci at accessible chromatin regions (in the case of multiple bins overlapping within a HERVH loci, the bin with the highest RPKM was considered).

## *Homo-Cynomolgus* cross-species comparative analysis

For this cross-species analysis, we selected 228 cells from human preimplantation blastocysts [2] (ICM, EPI, PrE, and TE) and 170 cells from *Cynomolgus* [86] (ICM, EPI, Hypoblast (PrE), and TE). Note that the *Cynomolgus* data predefined cell types for extraction and thus, as NCC cells were then not recognised, NCC cells could not be analysed in the cross-species analysis. For generating a cross-platform single-cell RNAseq dataset, counts were merged by gene name, and log2 TPM+1s were calculated. We redefined ICM, EPI, PrE, and TE cells using only those genes that were annotated in Refseq gene tracks of both human and *Cynomolgus* (similar in approach to [86]). This approach resulted in 16,222 individual genes that were merged into a single pool. We restricted the analysis to 11,053 orthologous genes that were expressed (Log2 TPM >1) in at least 5 cells out of the approximately 400 single cells in the merged ICM-EPI-PrE data frame of human and *Cynomolgus*. Variation due to batch effects was adjusted using COMBAT [87] from the R package sva. We checked the normalisation status by drawing PC biplots using various subsets of clustered genes to ensure that cells did not cluster on the basis of platform or species. We checked the validation of this analysis by visualising the selected gene expression (log TPM values) of conserved lineage markers across vertebrate blastocysts [86]. Notably, plots show a similar expression pattern of SPIC (ICM) NANOG, POU5F1, ICM/EPI, NODAL, GDF3 and PRDM14 (EPI), APOA1, GATA4 and COL4A1 (PrE) DLX3, STS and PGF (TE) in both human and macaque.

## Cross-species normalisation

Cross-species single-cell datasets were normalised using the recently published *Seurat Alignment* function [88]. First, we processed ICM, EPI, and PrE cells from either *Homo* or *Cynomolgus* using orthologous genes. We found similar expression patterns of conserved markers in the lineages of both species. We then detected variable genes in each of the datasets, using the *FindVariableGenes* function with default parameters of "Seurat." We then merged the log normalised and scaled datasets from both species into a single dataset. As defined in the manual, we used all unique genes from the intersection of the 2 variable gene sets from *Homo* and *Cynomolgus* dataset. We used this gene set as an input to canonical correlation analysis (CCA), and alignment was performed using canonical correlation vectors (CCVs) across datasets with the *AlignSubspace* function. This normalised set of cross-species data was used for downstream tSNE analysis. tSNE visualisation was done of the first 2 dimensions (tSNE1 and tSNE2). The dimension reduction method used was PCA with the *cca.aligned* parameter providing the first 2 dimensions.

## Cross-species markers

Cross-species markers in this study are based on orthologous genes of human and *Cynomolgus* ICM, EPI, and PrE lineages. Thus, we do not consider the species-specific genes that cannot, by definition, be expressed in both species (e.g., NANOGNB-ICM, ESRG-ICM/EPI, LINC00261-PrE). To determine species-specific expression of orthologous genes, we use the "percent of cells expressing a given gene" criteria: (i) genes that are expressed in >95% of the cells in a focal lineage of 1 species; (ii) in <5% of cells in the same lineage of the comparator species; and (iii) are significantly up-regulated in the focal lineage (*differentialGeneTEst* built-in function from "Monocle2") are considered to be cross-species markers of that focal lineage.

## Visualisation of reads

To visualise through IGV over Refseq genes (hg19), mapped reads (bam format) were converted into a signal file (bedGraph format) using STAR with parameters:—*runMode inputAlignmentsFromBAM—outWigType bedGraph—outWigStrand Unstranded*. Signals from ATAC-seq and ChIP-seq were obtained from MACS2, using the parameters: *-g hs -q 0.01 -B*. The conservation track was visualised through UCSC genome browser under net/chain alignment of given NHPs and merged beneath the IGV tracks.

## ChIA-PET and Hi-C analyses

These analyses were performed to address whether cohesin (SMC1) contributes to the pairing of LTR7-HERVH enhancer with promoters of the APOBEC3 gene cluster. SMC1 fastq reads of each mate was processed using the linkers [46]. After linker filtering, tags longer than 25 bp were considered for each mate. The resulting tags were mapped against the hg19 reference genome with the bowtie2 parameters:—sensitive-local. All unmapped reads (reads with MAPQ <10) and PCR duplicates were removed from the analysis. The aligned tags were paired using Picard and *samtools*, generating PETs (paired-end tags). The short read-length of the ChIA-PET dataset limits on the potential for proper coverage over repetitive REs (e.g., HERVH). Thus, it was necessary first to calculate the relative enrichment of the ChIA-PET signal in a 100 bp binned hg19 reference genome region, overlapping with an H3K27Ac region. The significantly enriched ChIA-PET signals in a particular bin (Observed/Expected >2 and an arbitrary threshold of 5 reads for each PET distinctly mapped on the genomic coordinates) were considered interactions.

## Defining Young and Old retroelements (REs) for comparative transcription analyses

As we expect recent RE introductions to be more likely to remain transpositionally active, or to have more recently been active, older ones having been transpositionally inactivated, we define Young RE families as integrated approximately during or after the split of human and chimpanzee, <7 MY [89]. Old REs, by contrast, are those that integrated before the split (>7 MY). Note that Young and Old are not defined by insertional mechanism. Indeed, the Young class includes the autonomous L1_Hs transposon that can mobilise SVA and certain ALU elements. ERVs feature in both Young (e.g., polymorphic HERVK-HML2) and Old (e.g., HERVH) groups. We only used the consensus of full-length REs to perform comparative analyses.

## Bulk RNAseq

HERVH$^{high}$ cells were generated by selecting cells tagged using GFP with an HERVH promoter as in [90]. Samples were prepared similarly to previous microarray analysis and subjected to bulk RNA sequencing. The RNAseq library preparation followed the Illumina TruSeq Stranded mRNA Sample Preparation Kit protocol on Illumina HiSeq machine with paired-end 151 cycles.

## Microscopy analyses on human embryo

Confocal analyses of LINE-1 ORF1p expression were analysed on a Zeiss LSM 710 confocal microscope using a previously described method [91]. Antibodies for the immunostaining: Rabbit anti LINE-1 ORF1p, 1:500, a generous gift of Dr. Oliver Weichenrieder (Max Planck, Germany). Secondary antibody: Alexa 488 Donkey anti Rabbit, 1:1,000 (Thermo). Mouse anti γ$^{+-}$H2AX, 1:200, clone 3F2 (Novus). Secondary antibody: Alexa 555 Donkey anti Mouse, 1:1,000 (Thermo). Rabbit anti cleaved caspase-3 (cl_Caspase3) (Cell Signalling, 9661S) was used at a 1:200 dilution. Secondary antibody: Alexa Fluor anti Rabbit 555 (1:1,000, Invitrogen). DAPI (Thermo) was used at 1:500. To assess the number of L1-ORF1p cytoplasmic foci in human blastocysts, every second image of a confocal stack was used at the height of the ICM. L1-ORF1p cytoplasmic foci were enhanced by applying a granulation filter with the same parameters to all images, and ORF1p cytoplasmic foci were counted manually. To assess L1-ORF1p$^{+}$- cl_Caspase3$^{+}$ double staining, 6 embryos were stained. To quantify the L1-ORF1p$^{+}$-Cl-csapase3$^{+}$ cells, 4/6 embryos were used.

## Plasmid constructs

We employed previously described shRNA constructs targeting HERVH [34,35] or scramble non-targeting control. shRNAs were cloned into the PB-H1 vector (modified from [34,35]) by BglII/ClaI restriction sites. Endotoxin-free plasmid preparations were performed using the NucleoBond Xtra (Macherey Nagel).

## Double-stranded DNA damage and early-stage apoptotic signal visualisation and in hESCs

To visualise double-stranded DNA damage and early-stage apoptotic conditions, we used γ-H2AX and cleaved cl_CASP3 immunostaining, respectively, followed by confocal microscopy. hESC_H9 human embryonic stem cells were cultured on coverslips, coated with matrigel (Corning) in Essential 8 media (Thermo Fisher) and transfected as described above. Starting 24 h after transfection, 10 μm capsaicin (Sigma) dissolved in ethanol was daily supplemented

to the growth media and withdrawn 24 h before fixation. Five days post transfection cells were fixed with 4% paraformaldehyde in 0.1% Triton X100 for permeabilization. Blocking was performed in 1% BSA and cells were stained with primary anti-γ-H2AX antibodies (1:2,000) (NovusBio) or anti-CASP3 antibodies (1:1,000) (Cell Signalling) overnight. For fluorescent visualisation, samples were incubated with secondary anti-mouse Alexa 647-conjugated antibodies (1:500) (BD Bioscience) or anti-rabbit Alexa 488-conjugated antibodies (1:500) (Thermo Fisher). Nuclei were stained with DAPI (0.5 μg/ml). For imaging, a Leica SP8 confocal microscope was used with similar settings for all samples, at least 2 technical replicates of each sample were acquired. Images were analysed with ImageJ 1.53a software (NIH). To quantify number of γ-H2AX and cleaved CASP3 foci, each image was analyzed with Huang, Yen, MaxEntropy, or Intermodes algorithm, depending on the original intensity of an acquired picture with watershed function to separate nuclei with similar intensity. Particles were analysed with "Particle Analysis" tool, for γ-H2AX signal: 5 to 60 pixels size, 0.4 to 1.00 circularity; for cleaved CASP3 signal: 50-Infinity pixels, 0.00 to 1.00 circularity; for DAPI signal: 50/100/150-Infinity pixels, 0.00 to 1.00 circularity. The number of γ-H2AX or cleaved CASP3 foci was normalised to the total number of nuclei per image of the corresponding sample. Experiment was done in 3 independent replicates for γ-H2AX gamma staining and in 2 for cleaved CASP3.

## Telomere length quantification in KD-HERVH cells

hESC_H1 human embryonic stem cells (WA01) (WiCell Research Institute) were grown on Nunc-treated plates (Thermo Fisher) coated with matrigel in mTeSR1 media (Stem Cell Technologies) and supplemented with primocin. Cells were passaged twice prior to transfections. hESC_H1 cells were treated with Accutase for 5 min at 37°C to achieve single cell suspension. One million hESC_H1 cells were transfected with 5 μg ESRG knock-down, scramble construct, or mock transfected using the Neon transfection system. ROCK inhibitor Y-27632 was added at 10 μm concentration into culturing media for the first 24 h. For quantifying telomere length, genomic DNA was isolated 48 h after the transfection using DNeasy Blood&Tissue Kit (Qiagen), and 1 ng of genomic DNA was analysed by quantitative PCR (qPCR) using Absolute Human Telomere Length Quantification qPCR Assay Kit (ScienCell Research Laboratories) according to manufacturer's instructions. Four individual experiments and 3 technical replicates were obtained per every transfection. Data are presented as telomere length +/− 0.16 kb per chromosome end and are normalised to mean of the distribution. Data were analysed with Kruskal–Wallis and Mann–Whitney $U$-test; samples were considered to be statistically different when $p < 0.05$.

## Ethics approval

The HERVH studies in human ESCs were performed under the allowance from the Robert Koch Institute AZ: 3.04.02/0119. The hESC line (H9) are permitted to be used in the study of "Untersuchung der HERVH-abgeleiteten regulation der Pluripotenz bei Menchen mit Hilfe der humanen embryonalen Stamzellen."

For the human embryo stainings, prior to the start of the project, the whole procedure was approved by local regulatory authorities and the Spanish National Embryo steering committee. Cryopreserved human embryos of the maximum quality were donated with informed consent by couples that had already undergone an IVF cycle. All extractions/manipulations were carried out in a GMP certified facility by certified embryologist in Banco Andaluz Celulas Madre, Granada, Spain.

## Supporting information

**S1 Fig. High-resolution dissection of human preimplantation development identifies an NCC population, excluded from early development.** Code to generate these figures is at https://doi.org/10.5281/zenodo.7925199. (A) Tracing the human embryonic development progression from zygote to blastocyst. PCA of cross-platform 1285+114 single-cell preimplantation transcriptome [1,2] using 1,583 MVGs. Developmental stages defined as in [2,40]. Note that this figure also demonstrates that any batch effects were adequately corrected prior to the analysis: instead of clustering due to different batches, single cells cluster on the basis of their embryonic stages (E3 with 8-cell, E4 with Morula, and E5-7 with blastocysts). (B) Feature plots based on tSNE plot from Fig 1A visualising the expression of selected lineage-specific markers, e.g., LEUTX (8-cell), FUT3 (morula), NANOG (EPI), BMP2 (ICM/PrE), DLX3 (pre-TE), CYP19A1 (TE). Colour intensity gradient indicates the expression of the marker gene (black, lower; coloured, higher). Each dot represents an individual cell. (C) Heatmap displaying the scaled expression (Log2 TPM values) of discriminative gene sets (AUC cutoff $\geq 0.90$) defining cell populations of morula ($n = 171$), EPI ($n = 52$), PrE ($n = 45$), and TE ($n = 99$) reported in Fig 1A. Heatmap colour scheme is based on Z-score distribution from $-2$ (light blue) to 2 (purple). Note: The majority of the markers agree with a previous study [2]. Here, we show the markers of EPI, PrE, and TE from E3–E7. (D) PAGA connectivity graph representation of raw single cell count datasets from E3 to E7 using default parameters of SCAMPY-PAGA. Coloured nodes represent subclusters of transcriptionally similar cells (threshold 0.1). The number of the cells forming a particular subcluster is reflected in the size of the circle. Numbers indicate cluster ID. Similarity between subclusters is indicated by connecting lines (thickness denotes the statistical measure of connectivity between clusters). Subclusters (circled) can form higher order categories that correspond to developmental stages of 8-cell, morula, and the blastocyst at E3, E4, and E5-7, respectively. Note that this analysis reproduces the basic tSNE analysis in Fig 1A as we notice that cell clusters marked by stage-specific markers, e.g., LEUTX-8 cell, FUT3-morula, and blastocyst markers (NODAL-EPI, APOA1-PrE, DLX3-TE) show interconnectivity. Additionally, the disconnected 3 subclusters at late E4 and E5 have distinct transcriptomes that represent the E5 cells of the NCC population, clusters 23 and 30 (Fig 1A) and a subcluster of E4 M2 Morula, cluster 11 (S5C Fig).
(PDF)

**S2 Fig. Noncommitted cell fate during human blastocyst formation.** Code to generate these figures is at doi.org/10.5281/zenodo.7925199. **(A)** DPT plot between Diffusion Component 1 and 2 (DC1 and DC2) illustrating 3 major states of E5 along pseudotemporal ordering. Bottom: pre-TE. top: ICM. Middle right: NCCs and cells are under progression. Cells coloured dark and golden yellow are representing higher and lower diffusion values. **(B)** The series of feature plots show the expression dynamics of individual genes in E5 blastocyst plotted on the DPT (A). Cells coloured dark and golden-yellow are representing higher and lower expression of respective genes, respectively. ICM (e.g., IL6R and SPIC) is characterised by the progressing cells enriched in EPI (e.g., NANOG and IFITM1) and PrE (e.g., BMP2 and PDGFRA). pre-TE population is identified by marker gene expression (e.g., DLX3 and GATA2). Note that NCCs do not express any lineage markers, but are marked by BIK and PPAP2C expression. **(C)** Pseudotime trajectory showing the ordering of E5 cells. Monocle2 visualisation of pre-TE, ICM, and NCC trajectories using the DDRTree algorithm. The top 2,000 DEGs are projected into a 2D space (right panel). **(D)** Monocle2 single cell trajectory analysis and ordered cells along an artificial temporal continuum using the top 1,000 DEGs across the data frame of E3–E5 cells. The transcriptome from each single cell represents a pseudotime point along an artificial time vector that denotes the progression from 8-cell to blastocyst via morula. Note: The artificial

time point progression agrees with the biological one. NCCs deviate on the trajectory. We show the trajectory in 6 facets, one for each cluster identified previously (Fig 1A). Colour codes as in (C). **(E)** Heatmap visualisation of scaled expression [log TPM (transcripts per million)] values of distinctive set of 1,000 genes (AUC cutoff >0.90) for each trajectory shown on (C). (AUC cutoff >0.90). Colour scheme is based on Z-score distribution from −2.5 to 2.5. Top margin colour bars highlight representative gene sets specific to the respective clusters. **(F)** tSNE clustering of E5 cells using approximately 1,000 MVGs reveals 3 distinct cell populations of pre-TE, ICM, and a previously undocumented cluster of NCCs. Each dot represents a single cell. (See also Fig 1E.)
(PDF)

**S3 Fig. Apoptotic cells in the human blastocyst.** Code to generate these figures is at doi.org/10.5281/zenodo.7925199. **(A)** UMAP clustering of E5 cells from an independent study [17] using approximately 1,000 MVGs reveals 4 distinct cell populations of pre-TE, ICM, Transitory, and NCCs (see the markers in (B)). Each dot represents a single cell. **(B)** RNA transcript intensity and density of differentially expressed markers of ICM, Transitory, pre-TE, and NCC clusters across the cell types obtained by the UMAP clustering (A). The dot colour scales from blue to red, corresponding to lower and higher expression, respectively. The size of the dot is directly proportional to the percentage of cells expressing the markers in a given cell type. **(C)** Reference based deconvolutions on bulk RNAseq (8-cell and ICM, GSE101571) and scRNAseq datasets using data from E3–E5 lineages. The stacked barplot shows the identified E3–E4 transcriptomes in the 8-cell bulk RNAseq, whereas the ICM-E5 sample has the identifiable lineages of Transitory, ICM, pre-TE, and NCC. We employ the single cell marker evidence to classify proportions of cell types. As these markers of NCC are apoptotic markers, we conclude that, in agreement with visualisations and scRNA data, approximately 20% of cells are NCC type. **(D)** Multiple violin plots visualise the density and distribution of expression (Log2 TPM values) of selected genes that are up-regulated in human NCC vs. EPI/PrE. The depicted genes (top candidates) are involved in regulating apoptotic pathways (KEGG: hsa04210, Gene Ontology GO:008219, GO:0012501, and GO:0006915) (Wilcoxon test, $p$-value $< 7.135 \times 10^{-6}$). **(E)** Feature plots based on tSNE plot (also shown on S5F Fig) demonstrating lineage-specific expression of apoptotic genes responding to DNA damage, e.g., TFEB [19] or TP53I13, the latter associated with telomere maintenance and genome stability [92]. Dots in yellow denote lower, whereas purple denote higher level of gene expression in a given single cell.
(PDF)

**S4 Fig. Transcriptional markers of committed lineages of the human blastocyst.** Code to generate these figures is at doi.org/10.5281/zenodo.7925199. **(A)** Representative confocal immunofluorescent images of human E5 blastocysts co-stained against NANOG (red), γ-H2AX (green), and DAPI (blue); brightfield, black-and-white panels. Cells in the blastocyst are stained either and exclusively with NANOG, representing the compacting cells of the ICM, or with γ-H2AX, representing damaged/dying cells. Note that the γ-H2AX$^{+}$ cells with disintegrating nucleus are distinct form the committed, pre-TE cells. Images of 3 different embryos are shown. Magnification is 40×. Venn diagram shows the numerical analysis of the immunofluorescent co-staining performed on the 3 independent embryos. See also Fig 2A. **(B)** Optimisation of immunostaining using NANOG (red) and BMP2 (green) antibodies in hESC_H9 cells. Magnification is 63×. **(C)** tSNE biplot visualises the human ICM ($n = 71$), PrE ($n = 45$), and EPI ($n = 52$) clusters using the most variable genes ($n = 532$) (see S3 Table for the full list of the markers). ICM is E5, EPI and PrE are E6-7. **(D)** Multiple feature plots based on tSNE plot from (C) displaying unsupervised identification of expression markers of ICM (NANOGBB, SPIC), ICM/EPI (NANOG, SCGB3A2), ICM/PrE (BMP2, PDGFRA), EPI

(NODAL, ATP12A), and PrE (COL4A1, APOA1) (see S3 Table for the full list of the markers). Dots in yellow/purple denote lower/higher expression in a given single cell, respectively. **(E)** Heatmap showing scaled expression (log TPM values) of distinctive marker gene sets defining EPI, PrE, and ICM. Genes specific to ICM include NANOGNB, PRSS3, and SPIC. Note: the progenitor markers that are homogeneously expressed in ICM, but also in EPI (e.g., NANOG, MT1X) or PrE (e.g., BMP2, PDGFRA). Note: Underlined genes are specific to human when compared with macaque. Colour scheme is based on *Z*-score distribution, from –2.5 (gold) to 2.5 (purple).
(PDF)

**S5 Fig. Activated "Hot" L1 _Hs elements in early human embryos.** Code to generate these figures is at doi.org/10.5281/zenodo.7925199. **(A)** A representative confocal immunofluorescent image of a human E5 blastocyst co-stained against L1-ORF1p (left 2 panels) (cytoplasmic, red) and DAPI (nuclear, blue; brightfield, black-and-white panel). Note: the scattered L1-ORF1p signal also in TE. Magnification is 40×. See also S1 Movie. Representative confocal immunofluorescent images of a human E5late blastocyst (right 4 panels) (brightfield, black-and-white panels) co-stained against POU5F1/OCT4 (nuclear, purple), L1-ORF1p (cytoplasmic, green), and DAPI (nuclear, blue). Note: The antagonistic expression of POU5F1/OCT4 and L1-ORF1p: POU5F1 expression stains the compacting cells, where L1-ORF1p expression is not detectable. L1-ORF1p+ cells are also detectable in TE. Magnification is 63×. **(B)** Regulation the expression of "hot" L1 elements in blastomeres and ICM. RNA-seq coverage shows the mapable reads; ATAC-seq/DNAse-seq coverage plots show the signals around the transcription start sites (TSSs) located at the left boundaries of "hot" L1 loci. X-axis, upstream 2 KB and downstream 8 KB regions from the left boundaries divided into 100 bins, each comprising 100 bps; Y-axis, normalised ATAC-seq signal (per million per 100 bp bin). Samples: Replicates (*n* = 2) of human 8-cell stage/morula embryo, human bulk-ICM. **(C)** Activity of 4 of the 6 "ultra-hot" L1 elements [10] in early human development (data source as in (B)). **(D)** tSNE unbiased clustering of E3–E4 cells using approximately 500 MVGs reveals 3 distinct cell populations (upper left panel). The 3 groups of cells are identified as 8 cells stage and 2 distinct populations of E4 embryo Morula 1 (M1) and Morula 2 (M2) (lower left panel). Each dot represents a single cell. On the right panels, we show feature plots based on tSNE plot demonstrating lineage-specific expression of LEUTX (8-cell marker [1,2,93]), HKDC1 (M1 marker, this study), GATA3 (M2 marker, this study, but pre-TE marker in [6,40,93]), and DNMT3B (M2 marker, this study, but blastocyst marker in [6,40]). Dots in purple denote higher level of gene expression in a given single cell.
(PDF)

**S6 Fig. Characterisation of the 2 human morula clusters.** Code to generate these figures is at doi.org/10.5281/zenodo.7925199. **(A)** Violin plots showing the expression distribution of TFCP2L1, ESRRB, and DNMT3L (naïve stem cell culture markers [94–96]) in 8-cell and morula, M1 and M2 stages. Adjusted *p*-values are obtained by Benjamini–Hochberg (BH) correction. Each point represents a single cell. **(B)** Dot plots of differentially expressed genes marking each cluster of cells shown in S5F Fig. The dot colour scales from blue to red, corresponding to lower and higher expression, respectively. The size of the dot is directly proportional to the percentage of cells expressing the markers in a given cell type. **(C)** tSNE clustering of E3–E5 cells from approximately 1,000 MVGs obtained using default parameters of "Seurat" package (upper panel) dissects 5 distinct groups of cells (upper left panel). The analysis of E5 blastocyst identifies pre-TE, ICM, and NCC. Transitory cells (T) are observed between ICM and NCC. Each dot represents an individual cell plotted on first 2 tSNE (lower left panel). On the right panels, we show feature plots based on the tSNE plot demonstrating lineage-specific

expression of LEUTX (8-cell marker [1,2,93]), FUT3 (morula marker [1,2]), HKDC1 [97] (morula M1 and NCC marker, this study), NANOG-BMP2 (ICM marker, this study), and DLX3 (pre-TE marker [2]). Dots in purple denote higher level of gene expression in a given single cell. **(D)** Cell-cycle scoring of E4 and E5 single cells with their cell-cycle characteristics. **(E)** Cell-cycle scoring of E4 and E5 single cells with their cell-cycle characteristics, as mainly inferred by the expression of key cell-cycle genes (ridge plots).
(PDF)

**S7 Fig. Characterisation of the HERVH high vs. low conditions in vitro.** Code to generate these figures is at doi.org/10.5281/zenodo.7925199. **(A)** Validation of HERVH depletion (KD-HERVH) using qPCR in hESC_H9s in 3 biological replicates. KD-Control (depleted GFP). **(B)** The effect of HERVH expression on lineage specification. Multiple jittered boxplots display the differential gene expression (DEG, Log2-fold change) of various blastocyst (ICM/PrE; PrE) lineage markers (red, HEVH$^{High}$ vs. HERVH$^{Low}$ cells; blue KD-HERVH vs. KD-GFP (control) in hESC_H1s). We show the top 5 markers. Solid dots represent the differential expression values of the individual genes in the boxplots. **(C)** The effect of HERVH expression on genome stability. Multiple jittered boxplots display the differentially expressed genes (DEG, Log2-fold change) of various genes of GO: *DNA damage responsive genes to induce apoptosis* and *TP53-mediated genes responsive to DNA damage* (dark red, HERVH$^{High}$ vs. HERVH$^{Low}$ cells; blue KD-HERVH vs. KD-GFP in hESC_H1s (control)). We show the top 5 to 6 markers. **(D)** The effect of HERVH expression on *telomere protection*. Jittered boxplot displays the DEGs (Log2-fold change) of GO category: *Positive regulation of telomere maintenance* (red, HERVH$^{High}$ vs. HERVH$^{Low}$ cells; blue KD-HERVH /hESCs_H1 vs. KD-GFP/hESC_H1s (control)). We show the top 5 markers. Solid dots represent the differential expression values of the individual genes in the boxplots. **(E)** Knocking down HERVH shortens the telomeres in hESC_H1. Telomere length was quantified from genomic DNA using qPCR 48 h after KD-HERVH, KD-Control (scrambled) and mock transfection. KD-HERVH significantly reduced telomere length ($P = 0.0317$ and $P = 0.0079$ compared with mock and scramble control, respectively; Mann–Whitney $U$-test). Data are presented as violin plots with telomere length $+/- 0.16$ kb per chromosome end and normalised to distribution in 5 biological repeats. **(F)** The LTR7/HERVH-int is fixed in the human population. (Upper panel) The structure of the genomic region upstream of the APOBEC3G locus in the human genome. (Lower panel) Jittered boxplots compare minor allele frequencies from the SNPs of LTR7-HERVH (annotated beneath the APOBECs gene structure) and the rest of the genomic region within the shown window (blue box beneath the HERVH structure). The frequency of the minor allele is significantly lower at the LTR7-HEVH-LTR7 locus in the human population compared with the neighbouring regions (data from 1,000 genome project [98]).
(PDF)

**S8 Fig. HERVH as a functional enhancer of a specific set of APOBEC3 genes in human pluripotency.** Code to generate these figures is at doi.org/10.5281/zenodo.7925199. **(A)** Line plot showing the expression of APOBEC3G in ICM, KD-GFP (Control) (2 replicates), HERVH [35] (2 replicates) in hESC_H1 and TE. Note that the relative expression of APOBEC3G is the highest in ICM, depleted in KD-HERVH and the lowest in TrEs. **(B)** Boxplot represents the distribution of relative average difference (at Log2 scale) of gene expression around HERVH loci. Single cells are pooled together, scaled, and averaged for the analysis. Only genes, neighbouring HERVH (10 KB window) and expressed at least in 10% of the cells are used for the analysis. Note: Up-regulated gene expression neighbouring HERVH (HERVH target) was specifically observed in ICM but neither in NCC nor in pre-TE transcriptomes. **(C)** Representative examples of chromatin loops (boxed, red-HERVH; blue-APOBECs) between

HERVH and the APOBEC3 genes in hESCs (zoom in, 5 KB resolution). Heatmaps show normalised counts of Hi-C reads between selected genomic loci pairs. Gene structure, phylogenetic conservation, SINE, LINE, LTR retroelements, and 3D interaction at the APOBEC3 (chr22) locus are shown. **(D)** Heatmap showing the scaled expression (Log2 FPKM) dynamics of the APOBEC3 family of genes during the preimplantation life of human embryo (left panel [44], right panel [1]). Blue denotes lower to no expression. Line plot beneath the heatmap shows the averaged raw expression (Log2 FPKM+1) of the APOBEC3 genes.
(PDF)

**S9 Fig. Young RE activity in human early embryogenesis.** Code to generate these figures is at doi.org/10.5281/zenodo.7925199. **(A)** The selection arena model in early human embryos. **(B)** Capsaicin treatment antagonises the effect of HERVH suppression in hESCs. HERVH suppression (KD) results in increased DNA damage and apoptosis in hESCs (visualised by γ-H2AX and cleaved cl_CASP3 immunostaining, respectively), but mitigated by capsaicin treatment. KD-HERVH_H9 and KD-Control_H9 (scramble). **(C)** Knocking down HERVH results in elevated DNA damage in KD-hESC_H9 (visualised by γ-H2AX immunostaining ($n = 3$)). However, inhibition of L1 activity by capsaicin reduces the DNA damage, visualised by immunofluorescent staining against γ-H2AX in KD-hESC_H9. KD-Control (scrambled). Error bar–standard error of the mean. Diluent, ethanol ($n = 3$). **(D)** Knocking down HERVH results in an enhanced apoptotic signal (cl_Caspase3). However, inhibition of L1 activity by capsaicin reduces the pre-apoptotic signal in KD-hESC_H9. Diluent, ethanol. KD-Control (scrambled). Error bar–standard error of the mean.
(PDF)

**S10 Fig. Single-cell transcriptome analysis of the mouse preimplantation embryo reveals no uncharacterised, apoptotic cell population (NCC).** Code to generate these figures is at doi.org/10.5281/zenodo.7925199. (A) Two-dimensional tSNE density cluster representation of 259 mouse single-cell preimplantation transcriptomes [99] using the 1,297 MVGs (analysed similarly to the human data). Dashed, curvy arrows indicate the progression of the embryogenesis. (B) PCA of morula and early blastocyst stages. The segregated clusters are based on the expression of MVGs. ICM (marked by NANOG), pre-TE (marked by CDX2), and transitory state (marked by NANOG/CDX2) are circled. (C) Heatmap displaying z-scores of up-regulated genes (red) during mouse preimplantation embryogenesis. Gene Ontology of genes enriched 2-fold in a particular developmental stage compared to the rest of the developmental stages. The data frame uses normalised mean TPM (calculated per mouse embryonic stage) 8-cell, morula and early/mid/late blastocysts. (D) Violin plot visualisation of the expression dynamic of selected pro-apoptotic marker genes in mouse embryogenesis. Note: There is no significant difference in pro-apoptotic marker gene expression following morula stage (NS, not significant).
(PDF)

**S1 Table. Transcriptional markers of morula and blastocyst lineages.**
(XLSX)

**S2 Table. The 235 marker genes as molecular signatures of the individual clusters at E5 blastocyst (ICM, NCC, and Pre-TE).**
(XLSX)

**S3 Table. Expression markers of ICM, ICM/EPI, ICM/PrE, EPI, and PrE in the human blastocyst.**
(XLSX)

**S4 Table. Phylogenetically conserved and species-specific expression markers of ICM, ICM/EPI, ICM/PrE, EPI, and PrE in the primate (e.g., human and macaque) blastocyst using orthologous set of genes.**
(XLSX)

**S1 Movie. Expression dynamics of L1-ORF1p (red) by immunofluorescence staining during the formation of the blastocyst.** Note that L1_HS_ORF1p accumulates in the cytoplasm of pre-TE and in the blastocoel cells (DAPI, blue). The movie contains all the Z stacks combined taken from a single embryo.
(MOV)

## Acknowledgments

We thank Tamás Raskó for help with the cellular assays.

## Author Contributions

**Conceptualization:** Manvendra Singh, Laurence D. Hurst, Zsuzsanna Izsvák.

**Data curation:** Manvendra Singh.

**Formal analysis:** Manvendra Singh, Aleksandra M. Kondrashkina, Vikas Bansal, Laurence D. Hurst.

**Funding acquisition:** Jose L. Garcia-Perez, Laurence D. Hurst, Zsuzsanna Izsvák.

**Investigation:** Manvendra Singh, Aleksandra M. Kondrashkina, Thomas J. Widmann, Christine Römer, Marta Garcia-Canadas, Laurence D. Hurst.

**Methodology:** Manvendra Singh, Aleksandra M. Kondrashkina, Jichang Wang, Jose L. Garcia-Perez, Laurence D. Hurst, Zsuzsanna Izsvák.

**Project administration:** Jose L. Garcia-Perez, Laurence D. Hurst, Zsuzsanna Izsvák.

**Resources:** Jose L. Cortes, Jichang Wang, Christine Römer, Jose L. Garcia-Perez.

**Software:** Manvendra Singh.

**Supervision:** Jose L. Garcia-Perez, Laurence D. Hurst, Zsuzsanna Izsvák.

**Visualization:** Manvendra Singh.

**Writing – original draft:** Manvendra Singh, Laurence D. Hurst, Zsuzsanna Izsvák.

**Writing – review & editing:** Manvendra Singh, Aleksandra M. Kondrashkina, Laurence D. Hurst, Zsuzsanna Izsvák.

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
