## [Editor Report · Decision Letter 0]

15 Nov 2022

Dear Laurence, 

Thank you for submitting your manuscript entitled "­Discovery of a novel human embryonic cell type allows definition of inner cell mass and suggests that the early embryo may be a clonal selection arena" for consideration as a Research Article by PLOS Biology. Please accept again my apologies for the long delay in sending you our decision.

We have now received the comments from the Academic Editor and we would like to share them with you to see what are your thoughts and potential responses. I am sure it is not unexpected that the Academic Editor could not give us a simple answer. Like us, s/he found the conclusions of the manuscript interesting, but also agreed with the criticisms raised by the reviewers and thought that some of them remain unaddressed. These are the main points:

1. The causality between TE expression and apoptosis is not clear, and this population could be a subset of TE/ICM cells that have high levels of the ectopic expression of TEs as a result of apoptosis and DNA damage and indeed this has been found for L1.

2. Do the ESC experiments mimic the NCC state? This is particularly problematic as they can serve as a demonstration of causal relationships, but given that the authors use standard and not naïve cell states, I am not sure how either the reporter or kd can be said to mimic ICM vs NCC states. There doesn't seem to be any analysis of the cell states created in these cultures, just an assessment of pluripotency markers.

3. After looking at the most comprehensive integrated human data, I can’t find evidence of the population described. This doesn’t mean that it’s not there, but it gives me concern. 

4. Is it so surprising that there is a population of apoptotic cells in the early primate embryo? I believe that there is an established high level of mutation in preimplantation development that is then sorted out with cell competition later.

As I mentioned, it would be very helpful if you could share your thoughts with us by email about these questions before deciding a way forward, but we do not feel that we can make a quick decision based on the assessment of the previous reviews. Given the history of your manuscript, we also understand you might not want to have a further round of review, which is what we can offer you at this stage, so please let us know if you prefer to withdraw the manuscript and try elsewhere. If you want to proceed, we would need to contact a trusted reviewer to obtain further advice, but one possibility we would like you to consider to increase the chances of publishing your manuscript is to convert it into a ‘Discovery Report.’ I am sure you are familiar with the format, but these papers describe novel and intriguing preliminary findings with the potential to lead to a significant new result for the field and this paper would be a good fit. As these articles have up to 4 figures, you would have to do a significant restructuring of your manuscript, but we do think that this would be beneficial for the narrative to focus on the main findings.

Before we can send your manuscript for review, we need you to complete your submission by providing the metadata that is required for full assessment. To this end, please login to Editorial Manager where you will find the paper in the 'Submissions Needing Revisions' folder on your homepage. Please click 'Revise Submission' from the Action Links and complete all additional questions in the submission questionnaire.

Once your full submission is complete, your paper will undergo a series of checks in preparation for peer review. After your manuscript has passed the checks it will be sent out for review. To provide the metadata for your submission, please Login to Editorial Manager (https://www.editorialmanager.com/pbiology).

Kind regards,

Ines

--

Ines Alvarez-Garcia, PhD

Senior Editor

PLOS Biology

---

## [Decision Letter · Decision Letter 1]

10 Feb 2023

Dear Laurence,

Thank you for your patience while your manuscript entitled "­­Discovery of a novel human embryonic cell type associated with activity of young transposable elements allows definition of inner cell mass" went through peer-review at PLOS Biology as a Discovery Report. Please also accept my apologies for the delay in providing you with our decision. Your manuscript has now been evaluated by the PLOS Biology editors, an Academic Editor with relevant expertise, and by two independent reviewers.

The reviews are attached below. As you will see, the reviewers find the conclusions of the manuscript interesting and worth considering them for publication. However, while Reviewer 2 is fully satisfied, Reviewer 1 thinks that some parts of the manuscript need to be rewritten to consider the current state of the art and to clarify some of the statements. 

In light of the reviews, we are pleased to offer you the opportunity to address the remaining points from this reviewer in a revision that we anticipate should not take you very long. We will then assess your revised manuscript and your response to the reviewers' comments with our Academic Editor aiming to avoid further rounds of peer-review, although might need to consult with the reviewers, depending on the nature of the revisions.

**IMPORTANT - SUBMITTING YOUR REVISION**

3. Resubmission Checklist

a) *PLOS Data Policy*

b) *Published Peer Review*

Sincerely,

Ines

--

Ines Alvarez-Garcia, PhD

Senior Editor

PLOS Biology

Reviewers' comments

Rev. 1:

In this manuscript, Singh et al, reanalyze published scRNAseq of human preimplantation embryos to distinguish ICM cells from their cells states at 5 d.p.f. This med them to identify pre-apoptotic cells. They then focus on ICM cells and study the activity of transposable elements. This raises an interesting discussion on how embryos are protected from transposable elements, relatively to their "age".

While of potential interest, this article suffers from a major narrative issue (as noted by previous reviewers). More efforts are needed to answer those previous comments. Find bellow points that could help improve the manuscript.

Major:

1/ I understand that this article might have been around for a long time, but it needs to be rewritten considering the state of the art at the moment it will be published, otherwise it will look odd. Maybe start by saying you are performing a complementary approach to Radley et al in order to define human ICM and clearly identify the "other" cells that are generally ignored.

2/ The name "NCC" does not make sense if those cells are undergoing apoptosis sooner or later. Instead, I would insist on the fact that an important % of cells in human embryos have failed specification and are dying (you could speculate about their potential aneuploidy in the discussion). This is supported by the nice deconvolved bulk RNAseq, and it is novel.

Authors could show was the annotation of the "other" cells in Radley et al or Meistermann et al, for instance, to show that it is important to identify those cells in order to clean up subsequent analyses.

3/ The abstract/introduction suffer from erroneous statements, truly impairing credibility of the work. For instance:

"On a single-cell level, there are uncharacterised cell types": what does that mean?

"Perhaps most importantly, we still have no clear definition of the inner cell mass (ICM), the transitory group of pluripotent cells from which embryos are derived and hence of interest in regenerative medicine" ICM are the progenitors of EPI which are pluripotent. Embryos also refers to the placenta, you could use foetus, alternatively. Main application of human preimplantation development is IVF.

4/ The quality of IFs in fig2 is really sub-standard.

5/ Fig 1A: the "mural" and "polar" cluster are odd (really small) compared to previous studies such as Meisterman et al. The big E6-E7 cluster looks like mural TE.

Minor:

1/ it is better to use d.p.f. rather than "E" for human embryos.

2/ what is the RNA velocity of "NCC"

3/ you seem to have inverted TE (Trophectoderm) and TrE (Transposable elements)?

4/ your yellow-violet scale in heatmaps is inverted: the warm color (yellow) should be for high expression.

Rev. 2:

In this manuscript, Singh and co-workers seek to scrutinise the identity and fate of cells within the human inner cell mass, exploring the role of transposable elements in particular. The authors search published single cell multi-omics data obtained from human and other primate embryos, perform immunofluorescence and make use of human embryonic stem cells as an in vitro model. They uncover populations of cells associated with cell death and others appearing uncommitted, which they call 'non-committed cells (NCCs)'. They use published data to characterise pre-epiblast stage cells, revealing that NCCs are represented only up until ~E5, and tend to express pro-apoptotic genes. To control against the risk that handling for single cell isolation causes cell death they processed some morulae and ICMs by bulk sequencing and found the same proportion of dead cells as encountered using separated cells, which is an important, but often ignored validation. Further analysis of NCCs demonstrated that they express BCL2-Interacting Killer (BIK) and other genes that foretell programmed cell death. They also show markers of DNA damage. By immunofluorescence the authors found mutual exclusivity for NANOG and the death mark, gamma H2AX, implying that markers of apoptosis are not part of the ICM repertoire, as previously thought (and previously reported in the ICMs of developing mouse embryos), but transitory and excluded after E5. Early (E5) ICMs exhibit 3 clusters: Trophoblast, EPI/PE precursors and NCCs. To monitor how the lineages resolve during ICM maturation, embryos were stained for NANOG and BMP2. Several 'ultra-hot' Line 1 ORFs encoding ORF1P elements are observed in E5-6 embryos, but do not overlap with OCT4, therefore assigned TrE identity, also consistent with position within the embryo. This population of TrE cells co-stains with Caspase 3, so probably represents cells displaced to the extra-embryonic tissues, as previously suggested. ICM and NCC have distinct families of TEs (Old versus Young, respectively). One of the Old ones (HERVH-int) is considered to be a binding site for pluripotency transcription factors. The authors create a model for the pre-differentiating ICM using hESCs. High HERVH cells (marked with GFP expression tag) correspond with ICM and EPI pluripotency markers. TE-restricting factors tend to be exclusively in ICM. The 'Young' TEs are suppressed by APOBEC3, activated by LTR7/HERVH in ICM.

The data presented in this study provide further evidence for sequential rather that synchronous divergence of the founder lineages of the blastocyst, which was first published in a massive, high profile single cell sequencing study, but subsequently refuted using more refined analysis. One major highlight of the present manuscript is the detailed and insightful discussion, using the data generated for this study and a very comprehensive review of the literature, to begin to explain the mechanism by which unwanted/aneuploid cells may be diverted from the emerging epiblast.

Finally, the authors propose that TEs may regulate transition between developmental stages in human as well as other mammals.

The authors have made every effort to address comments from reviewers when this work was under consideration with a different publisher, strengthening the manuscript considerably and I have nothing further to criticise.

---

## [Editor Report · Decision Letter 2]

28 Mar 2023

Dear Laurence,

Thank you for your patience while we considered a new revised version of your manuscript entitled "­­Discovery of a novel human embryonic cell type associated with activity of young transposable elements allows definition of inner cell mass" for publication as a Discovery Report at PLOS Biology. This revised version has been evaluated by the PLOS Biology editors and by the Academic Editor.

Based on our Academic Editor's assessment of your revision, we are likely to accept this manuscript for publication, provided you satisfactorily address several remaining points as following:

1) We think Reviewer 1 is correct and that the term NCC cannot be used because these are not "non-committed cells," they are young or active transposable element expressing cells. Non committed has a very specific developmental implications, cells that have begun to differentiate, but retain the capacity to change their trajectory. In the preimplantation development field, it is used to refer to the capacity of cells identified based on marker expression that have the capacity to differentiate into different lineages in grafting or injection experiments. Thus, I am afraid it would not be correct to use it in the manuscript and must be changed.

2) These cells should be placed in context of the '8-cell state,' that expresses TEs (transposable elements). Presumably cells go from ZGA and ERV expression (with all TEs expressed) to ICM vs NCC (as they are currently called). Thus please clarify if your pre-TE state is preTE or preTrE and how does this relate to the 8C state.

3) We agree with Reviewer 1 that you should use the standard embryological abbreviations to avoid confusing the readers: TE for trophoectoderm, rather than TrE, and PrE for primitive endoderm. 

4) Regarding the naive issue, it seems that the paper contains experiments on primed cells and also a demonstration in the supplement figures with naive. Thus, this must be discussed to consider why primed cells should have this population, and the major data is from primed ESCs.

5) While the text has significantly improved, it is still hard to see the ICM as undefined. It is long been known to express multipe lineage determinants at the same time, e.g. Nanog and GATA6.

Please also make sure to address the data and other policy-related requests stated below.

We expect to receive your revised manuscript within two weeks. 

*Published Peer Review History*

*Press*

Sincerely,

Ines

Ines Alvarez-Garcia, PhD

Senior Editor

PLOS Biology

DATA POLICY:

Fig. 1B, C, E; Fig. 2F; Fig. 3A, C-F; Fig. 4A-D; Fig. S1A-D; Fig. S2A-C, E, F; Fig. S3A, C-E; Fig. S4C-E; Fig. S5B, D; Fig. S6A, C-E; Fig. S7A-F; Fig. S8A-D; Fig. S9C, D and Fig. S10A-D

We note that some of these are shown in Tables S2-S4, thus please indicate this in the figure legend along with where the other data are located.

Please also ensure your supplemental data file/s has a legend.

We require the original, uncropped and minimally adjusted images supporting all blot and gel results reported in an article's figures or Supporting Information files. We will require these files before a manuscript can be accepted so please prepare and upload them now. Please carefully read our guidelines for how to prepare and upload this data: https://journals.plos.org/plosbiology/s/figures#loc-blot-and-gel-reporting-requirements

DATA NOT SHOWN?

---

## [Editor Report · Decision Letter 3]

12 May 2023

Dear Laurence,

Thank you for the submission of your revised Discovery Report entitled "A new human embryonic cell type associated with activity of young transposable elements allows definition of the inner cell mass" for publication in PLOS Biology. On behalf of my colleagues and the Academic Editor, Josh Brickman, I am delighted to say that we can in principle accept your manuscript for publication, provided you address any remaining formatting and reporting issues. These will be detailed in an email you should receive within 2-3 business days from our colleagues in the journal operations team; no action is required from you until then. Please note that we will not be able to formally accept your manuscript and schedule it for publication until you have completed any requested changes.

PRESS

Sincerely, 

Ines

--

Ines Alvarez-Garcia, PhD

Senior Editor

PLOS Biology
